# GENERATIVE MODEL VIA QUANTILE ASSIGNMENT

## ABSTRACT

Deep Generative models (DGMs) play two central roles in modern machine learning: (i) producing new information (e.g., image synthesis, data augmentation, and creative content generation) and (ii) reducing dimensionality. Yet, DGMs' versatility must confront training difficulty. While deep neural networks (DNNs) are a natural choice for parameterizing generators, there is no universally reliable method for learning compact latent representations. As a compromise, current approaches rely on introducing an additional DNN: (i) variational autoencoders (VAEs), which map data into latent variables through an encoder, and (ii) generative adversarial networks (GANs), which employ a discriminator in an adversarial framework. Training these auxiliary networks jointly, however, introduces instability, increases computational cost, and may lead to optimization failures such as mode collapse. Here, to address these challenges, we introduce NeuroSQL and a new generative paradigm, a DGM that learns low-dimensional latent representations without an encoder. Specifically, NeuroSQL learns the latent variables implicitly by solving a linear assignment problem and then passes the latent information to a unique generator. To demonstrate NeuroSQL's efficacy, we benchmark its performance against GANs, VAEs, and a budget-matched diffusion baseline on four independent datasets on handwritten digits (MNIST), faces from the CelebFaces Attributes Dataset (CelebA), animal faces from Animal Faces HQ (AFHQ), and brain images from the Open Access Series of Imaging Studies (OASIS). Compared to VAEs, GANs, and diffusion models: (1) in terms of image quality, NeuroSQL achieves overall lower mean pixel distance between synthetic and authentic images and stronger perceptual/structural fidelity, under the same computational setting; (2) computationally, NeuroSQL requires the least amount of training time; and (3) practically, NeuroSQL provides an effective solution for generating synthetic data when there are limited training data (e.g., data with a higher-dimensional feature space than the sample size). Taken together, by embracing quantile assignment rather than an encoder, NeuroSQL provides a fast, stable, and robust way to generate synthetic data with minimal information loss.

## 1 INTRODUCTION

Deep generative models (DGMs) have become a cornerstone of machine learning and have made meaningful contributions to image synthesis, data augmentation, and creative content generation. Over the past decade, they have become ubiquitous in scientific fields, such as genomics and neuroimaging, to handle complex data analysis tasks, including data interpretation, decoding, and generating intricate datasets.

A large share of these advances has been powered by variational autoencoders (VAEs; Kingma & Welling, 2014) and generative adversarial networks (GANs; Goodfellow et al., 2014), which remain the two dominant approaches for generative modeling from lower-dimensional latent spaces. Both frameworks adopt a common strategy: pairing a generator with a complementary deep neural network (DNN). In VAEs, an encoder learns to map observations to latent variables, whereas in GANs, a discriminator provides adversarial feedback to indirectly train the generator.

Despite their promise, training DGMs remains a complex and challenging task. These approaches introduce both conceptual and practical limitations. Conceptually, there is no guarantee that an encoder or discriminator, viewed as a continuous mapping that a DNN can approximate, exists in the first place. For example, if the data manifold is non-smooth, DNN approximations may fail

to converge uniformly. While the generator is a function of the latent variable, which can have (much) lower dimension than the observations, both the encoder and discriminator are functions of the (potentially large) data itself, and thus may encounter a curse of dimensionality (Stone, 1985). While DNNs are believed to enjoy fast rates of convergence (Schmidt-Hieber (2020)) or can, under favorable conditions, mitigate the curse of dimensionality (see e.g. Suzuki (2019)), these properties are derived under strict assumptions (Golestaneh et al., 2025). These limitations are reflected in theoretical works on VAEs and GANs, which often sidestep these issues. For instance, Chae et al. (2023) analyzes direct optimization of the likelihood, without introducing the encoder, which is unfeasible in practice. Biau et al. (2020) assumes from the outset that the discriminator belongs to a parametric function class. Ideally, however, assumptions should be placed only on the true objects of interest: the generator and the latent space.

In practice, training auxiliary networks is often more unstable and data-intensive than training the generator itself. GANs, in particular, are prone to convergence failures such as mode collapse (Mescheder et al., 2018). VAEs often suffer from blurred reconstructions due to their variational approximations and pixel-wise reconstruction losses, such as the mean squared error (MSE). More broadly, jointly learning multiple deep networks tends to exacerbate challenges of sample complexity, computational cost, and training instability. Furthermore, training DGMs requires large sample sizes, and when the data is large or high-dimensional (compared to the sample size), they may behave unfavorably. To address these issues, we consider quantile assignment in Section 3.

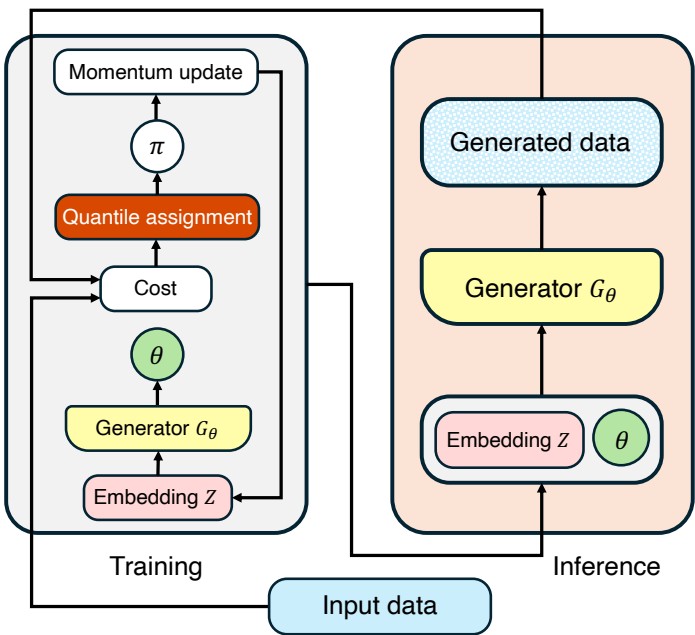

Figure 1: **A schematic representation of the NEUROSQL architecture.** Left: Algorithm for optimizing latent embeddings and generator parameters $\theta$. Right: Data flow in NeuroSQL for image synthesis. From left to right, input data enters the NeuroSQL; the model learns the embedding and parameter $\theta$; the embedding enters a generator parameterized by $\theta$; NeuroSQL outputs generated data. Here, "cost" on the left panel refers to cost matrix entries: $C_{i,k} = \ell\big(X_i, G_\theta(Q_k)\big)$, where $\ell(x, \hat{x}) = \frac{1}{2}(1 - \text{SSIM}(x, \hat{x})) + \frac{1}{2}\|x - \hat{x}\|_1$ is the same loss used to train the generator. "Momentum update" indicates: $\hat{Z}i^{(t)} \leftarrow \rho Q\pi^{(t)}(i) + (1 - \rho)\hat{Z}_i^{(t-1)}$, which smooths the latent code trajectory and stabilizes training.

Among generative models lacking a well-defined latent space, diffusion models employ a likelihood-based progressive denoising procedure (Ho et al., 2020; Dhariwal & Nichol, 2021; Rombach et al., 2022), which supports stable training and high-quality output generation. Nevertheless, Diffusion models often suffer from slow sampling and high computational and memory costs (Song et al., 2021; Nichol & Dhariwal, 2021; Lu et al., 2022). Theoretically, it was found that diffusion models

suffer from the curse of dimensionality unless the observations are generated by a linear factor structure (Oko et al., 2023).

Here, to address these issues, we introduce NEUROSQL, a new latent variable DGM that fundamentally differs from existing approaches relying on encoders, discriminators, or complex sampling procedures. NEUROSQL approximates the latent space using a permutation of quantiles, defined through optimal transport, and the latent variables implicitly by solving a linear assignment problem. As a result, NEUROSQL trains a unique DNN for the generator, alongside an assignment procedure. NEUROSQL is fast, stable, resource-friendly, and generally generates high image quality given the same computational resources. Experimental tasks on generating digits, human faces, animal faces, and brain imaging data suggest NEUROSQL provides an effective approach for generating synthetic data when there are limited training data (e.g., neuroimaging data with a higher-dimensional feature space than the sample size). Finally, by integrating Statistical Quantile Learning (SQL; Bodelet et al., 2025), which was developed for additive latent variable models, and a generator, NEUROSQL achieves a marriage between generative models and the blessing of dimensionality.

## 2 RELATED WORK

**GANs and VAEs.** There are some recent attempts to improve GAN, notably the StyleGAN, (Karras et al., 2019), which transformed controllable image generation through style-based architectures, as well as StyleGAN2 and StyleGAN3, which are considered to be state-of-the-art for several benchmarking datasets (Karras et al., 2020; 2021). In parallel, (Brock et al., 2019) demonstrated the importance of scale with BigGAN, while (Kynkäänniemi et al., 2019) proposed improved training techniques for large-scale GANs. While these approaches are beginning to address artifacts and improve training dynamics, they nonetheless suffer from training instability and mode collapse as they are naturally GANs and, therefore, all rely on the (minimax) adversarial training. Most recently, however, the adversarial objective has found renewed utility in accelerating diffusion models, where approaches like Adversarial Diffusion Distillation (ADD) employ discriminators to enable single-step high-fidelity sampling (Sauer et al., 2024). To improve reconstruction quality, Vector Quantized VAEs (VQ-VAE) discretise the latent space (Razavi et al., 2019), and VQ-VAE-2, a follow-up work, extends the original VQ-VAE by introducing hierarchical modeling (Child, 2021), thereby achieving high-resolution image generation. Combining the benefits of VQ-VAE with adversarial training, VQGAN further enhanced perceptual quality (Esser et al., 2021). Building on these quantized latent spaces, recent work in Visual Autoregressive Modeling (VAR) has redefined the paradigm from pixel-wise raster scanning to coarse-to-fine next-scale prediction, reportedly surpassing diffusion transformers in efficiency and scalability (Tian et al., 2024).

**Diffusion Models.** The next phase of generative AI emerged with the introduction of denoising diffusion probabilistic models (DDPM) (Ho et al., 2020), commonly referred to as diffusion models. An extension of DDPM was proposed by (Song et al., 2021), providing theoretical foundations for score-based generative models. Thanks to diffusion models' progressive denoising procedure, they have achieved most of the current state-of-the-art results in image and video generation. Recent advances in diffusion modeling include improved sampling efficiency (Song et al., 2021; Lu et al., 2022), conditional generation (Dhariwal & Nichol, 2021), and classifier-free guidance (Ho & Salimans, 2022). Although diffusion models achieve high-quality data generation, they are computationally intensive and require lengthy training time. A few works aim to reduce computational cost, such as Latent Diffusion Models (LDMs) (Rombach et al., 2022) and faster sampling (Song et al., 2023; Luo et al., 2023). As one of the most actively researched fields, improvements are constantly being proposed (Peebles & Xie, 2023; Chen et al., 2024). While most of the newer versions of diffusion models excel at generating high-quality data, and despite plentiful attempts to reduce the computational and time cost, the sequential denoising steps have remained a bottleneck.

## 3 METHODOLOGY

### 3.1 THE MODEL

In what follows, we first state the model we aim to learn. We consider a dataset $\mathcal{D}$ composed of $n$ independent copies of $p$-dimensional random vectors, $\mathcal{D} = \{\boldsymbol{X}_1, \ldots, \boldsymbol{X}_n\}$. We assume that the

data are driven by unobserved continuous latent variables $\boldsymbol{Z}_i \in \mathcal{Z} \subseteq \mathbb{R}^d$. Specifically, we consider a probabilistic generative model, which models the observations:

$$\boldsymbol{X}_i = \boldsymbol{G}(\boldsymbol{Z}_i) + \boldsymbol{\epsilon}_i, \tag{1}$$

where $\boldsymbol{G} : \mathcal{Z} \to \mathbb{R}^p$ is an unknown generator, and $\boldsymbol{\epsilon}_i$ are independent random errors. We assume that the latent variables follow a known continuous distribution $P_Z$. The latent dimension is typically (much) smaller than the ambient dimension ($d << p$) to enable dimensionality reduction. This, therefore, contrasts with a flow-based generative model, where the generator must be invertible. As DGMs are not identifiable, one can select any distribution as long as it has a continuous cumulative distribution function. Regarding the prior distribution, it is common to use either the standard normal distribution, $\mathcal{N}(\boldsymbol{0}, \boldsymbol{I})$, or a Uniform distribution. We approximate the generator using deep neural networks (DNN), that is $\boldsymbol{G} \approx \boldsymbol{G_\theta}$, where:

$$\boldsymbol{G_\theta} \in \{W_L \circ \sigma \circ W_{L-1} \circ \sigma \cdots \circ \sigma \circ W_1\},$$

and $W_l$ denote affine transformations and $\sigma$ is an activation function. The vector $\boldsymbol{\theta} \in \Theta$ contains the DNN parameters. We aim to learn both the generator parameter $\boldsymbol{\theta}$ and the latent variables $\boldsymbol{Z}_i$.

## 3.2 Latent space approximation and loss function

Our aim is to learn both the generator and the latent variables. In the context of deep generative models, computing and optimising the likelihood is feasible only for simple frameworks, such as linear factor analysis. In this section, we describe our estimation method with inspiration from the sieve method. The sieve method replaces an intractable optimization over the full parameter space with tractable problems on a growing sequence of simpler, typically finite-dimensional, subspaces, called sieves, that are dense in the parameter space (see e.g. Chen (2007) for a review).

Our aim is, therefore, to build an approximation for the latent space in order to obtain a tractable problem. The construction of this approximate latent space relies on partitioning the latent space into $n$ regions. Specifically, we consider quantiles denoted by $\boldsymbol{Q}_1^n, \ldots, \boldsymbol{Q}_n^n \in \mathcal{Z} \subseteq \mathbb{R}^d$, and let $\boldsymbol{Z} := (\boldsymbol{Z}_1, \ldots, \boldsymbol{Z}_n)'$ and $\boldsymbol{Q}^n := (\boldsymbol{Q}_1^n, \ldots, \boldsymbol{Q}_n^n)'$ be the $n \times d$ matrices of the latent variables and quantiles respectively. The basic idea of NEUROSQL is based on the fact that there exists a permutation $\pi$ such that the latent variable can be approximated by:

$$\boldsymbol{Z} \approx \boldsymbol{Q}_\pi^n, \tag{2}$$

where for a matrix (or column vector) $\boldsymbol{A}$ and a permutation $\pi$, we denote by $\boldsymbol{A}_\pi$ a permutation of the rows of $\boldsymbol{A}$. In contrast to the real line, there is no canonical ordering of $\mathbb{R}^d$, and thus there is no universal agreed-upon definition of "multivariate quantiles". Therefore, for clarity of exposition, we formalize here the approximation 2 for the univariate case, and will continue directly with the discussion of our method. We formalize the construction of $d$-dimensional quantiles in Section 3.3.

For $d = 1$, we select $(n + 1)$- quantiles, i.e. $\boldsymbol{Q}_i^n = F^{-1}(\frac{i}{n+1})$, $i = 1, \ldots, n$, where $F$ denotes the cumulative distribution function associated to $P_Z$. It can be shown using the delta method (see, e.g.,van der Vaart 2000) that:

$$|\boldsymbol{Z}_{(i)} - \boldsymbol{Q}_i^n| = \mathcal{O}_p\left(\frac{1}{\sqrt{n}}\right), \tag{3}$$

where $\boldsymbol{Z}_{(i)}$ denotes the order statistics of the latent variables. Note that the $\boldsymbol{Z}_{(i)}$'s are distinct almost surely and are thus, almost surely, a permutation of the $\boldsymbol{Z}_i$'s. We can thus express the approximation error by:

$$\min_{\pi \in S_n} \frac{1}{n} \sum_{i=1}^n |\boldsymbol{Z}_i - \boldsymbol{Q}_{\pi(i)}^n|^2 = \mathcal{O}_p\left(\frac{1}{n}\right), \tag{4}$$

where $S_n$ denotes the symmetric group of order $n$. Therefore, one can obtain an approximation of the latent variables by simply learning permutations. Intuitively, the set of all permuted quantiles, $\{Q_\pi : \pi \in S_n\}$, can be thought of as a "probabilistic sieve" space for the latent space and 4 as the sieve error, which is, in the language of sieve theory, the error that one makes by replacing the "true parameter" (in our case the $\boldsymbol{Z}_i$'s) by the sieve solution, or the closest element of the sieve space (the optimal permutation of the quantiles). Equation 4 tells that this approximation error vanishes asymptotically with $n \to \infty$, which is a desirable feature.

Leveraging this approximation, we consider the following criterion:

$$\mathcal{L}(\boldsymbol{\theta}, \pi) = \frac{1}{n} \sum_{i=1}^{n} \ell\left(\boldsymbol{X}_i, \boldsymbol{G}_{\boldsymbol{\theta}}\left(\boldsymbol{Q}_{\pi(i)}^n\right)\right),$$

for some loss function $\ell : \mathbb{R}^p \times \mathbb{R}^p \to \mathbb{R}_+$. We then define NEUROSQL as the solution of:

$$(\hat{\boldsymbol{\theta}}, \hat{\pi}) = \underset{\boldsymbol{\theta} \in \Theta, \, \pi \in S_n}{\arg\min} \; \mathcal{L}(\boldsymbol{\theta}, \pi) + \lambda\, \mathcal{R}(\boldsymbol{\theta}), \tag{5}$$

where $\mathcal{R}(\boldsymbol{\theta})$ is a regularizer controlling the complexity of $\boldsymbol{G}_{\boldsymbol{\theta}}$ and $\lambda > 0$ is some tuning parameter. With the solution $\hat{\pi}$, we then obtain the estimator of the latent variable as $\hat{\boldsymbol{Z}} = \boldsymbol{Q}_{\hat{\pi}}$.

### 3.3 MULTIVARIATE QUANTILES ($d > 1$)

Building on the univariate case, we extend to $d > 1$ in this subsection. Specifically, we discuss approximating the latent space in higher dimensions and show that the approximation error vanishes as $n \to \infty$. As there is no canonical ordering when $d > 1$, several methods have been developed to construct "multivariate quantiles". We concentrate on recent developments that leverage the optimal transport approach (Hallin, 2022; Chernozhukov et al., 2017; Hallin et al., 2021; Ghosal & Sen, 2022), which offer a conceptually clean and practical way to define quantiles in multivariate dimensions.

To build those multivariate quantiles, we use a regular grid, $\boldsymbol{U}_1, \boldsymbol{U}_2, \ldots, \boldsymbol{U}_n \in \mathcal{U}_d$ where $\mathcal{U}_d := \{\boldsymbol{z} \in \mathbb{R}^d : \|\boldsymbol{z}\| < 1\}$ is the unit ball. This grid does not have to be perfectly regular, which is in general not possible for $d \geq 3$. We require only that the discrete distribution with probability $1/n$ at each grid point converges weakly to the uniform distribution over $\mathcal{U}_d$. In practice, it is suitable to select a grid with a low discrepancy in order to obtain fast convergence rates.

In the case that the latent variables are uniformly distributed over $\mathcal{U}_d$, we simply define the multivariate quantiles as $\boldsymbol{Q}_i^n := \boldsymbol{U}_i$. For more general distributions $F$, we define the multivariate quantiles as $\boldsymbol{Q}_i^n := F_\pm^{-1}(U_i)$, where $F_\pm$ denotes the center-outward distribution function. Specifically, $F_\pm$ is the unique gradient of a convex function pushing $P_Z$ forward to the uniform distribution over the unit ball. We refer to Hallin et al. (2021) and Hallin (2022) for a detailed explanation. To avoid unnecessary technicalities, we will assume here that the distribution $P_Z$ is uniform over $\mathcal{U}_d$, yielding $\boldsymbol{Q}_i^n = \boldsymbol{U}_i$. We note that this is without loss of generality, as the latent distribution of (deep) generative models is not identifiable and should be selected (we refer to Bodelet et al., 2025 for a discussion).

The following proposition shows that the approximation error also vanishes asymptotically in the multivariate case.

**Proposition 1.** *Assume that the discrete distribution with probability $1/n$ at each grid point $\boldsymbol{U}_1, \boldsymbol{U}_2, \ldots, \boldsymbol{U}_n \in \mathcal{U}_d$ converges weakly to the uniform distribution over $\mathcal{U}_d$. Then the following holds:*

$$\min_{\pi \in S_n} \|\boldsymbol{Z} - \boldsymbol{Q}_\pi^n\|^2 \to 0, \; a.s.$$

*as $n \to \infty$.*

The proof of Proposition 1 follows from Theorem 2.4 in Hallin et al. (2021) and can be found in the Appendinx L.

### 3.4 COMPUTATIONAL ALGORITHM

We note that, for fixed $\boldsymbol{\theta}$, the inner problem in Eq 5 can be formulated as a *linear assignment problem*:

$$\min_{\pi \in S_n} \frac{1}{n} \sum_{i=1}^{n} C_{i,\pi(i)}, \qquad C_{i,k} := \ell\big(\boldsymbol{X}_i, \boldsymbol{G}_{\boldsymbol{\theta}}(\boldsymbol{Q}_k)\big),$$

where $\boldsymbol{C} := (C_{i,k})_{1 \leq i,k \leq n}$ is the cost matrix. This is solvable *exactly* by the Hungarian method in $O(n^3)$ time (Kuhn, 1955). Furthermore, for fixed $\pi$, equation 5 reduces to standard supervised regressions of $\boldsymbol{X}$ on assigned codes $\{z_{\pi(i)}\}$. We therefore propose to solve equation 5 alternatively:

(i) Given a permutation $\pi$, minimize the loss function with respect to $\boldsymbol{\theta}$ (Generator step); (ii) Given $\boldsymbol{\theta}$, we solve the linear assignment matching problem (via Hungarian or Greedy method). We iterate (i) and (ii) until convergence. Furthermore, we introduce a momentum update after each assignment, that is $\widehat{z}^{(t)} \leftarrow \rho\, z_{\pi^{(t)}(i)} + (1 - \rho)\, \widehat{z}^{(t-1)}$ for some $0 \leq \rho < 1$, in order to stabilize training. The exact steps are detailed in Algorithm 1.

---

**Algorithm 1** NEUROSQL (full-batch assignment)

1: **Input:** data $\{\boldsymbol{X}_i\}_{i=1}^n$, prior $P_Z$, lattice $\boldsymbol{Q}^n$, outer iters $T$, momentum $\rho \in [0, 1)$, $\lambda > 0$
2: Initialize $\pi^{(0)}$ (e.g., random or PCA-sorted); set $\widehat{\boldsymbol{Z}}^{(0)} = \boldsymbol{Q}^n_{\pi^{(0)}}$
3: **for** $t = 1, \dots, T$ **do**
4:    **Decoder step:** $\boldsymbol{\theta}^{(t)} \in \arg\min_{\boldsymbol{\theta}} \dfrac{1}{n} \sum_{i=1}^n \ell\big(\boldsymbol{X}_i, \boldsymbol{G_\theta}(\widehat{\boldsymbol{Z}}_i^{(t-1)})\big) + \lambda \mathcal{R}(\boldsymbol{\theta})$    *(implemented via*
   *AdamW with standard NN init on first call)*
5:    **Cost matrix:** $C_{i,k}^{(t)} \leftarrow \ell\big(\boldsymbol{X}_i, \boldsymbol{G}_{\boldsymbol{\theta}^{(t)}}(\boldsymbol{Q}_k)\big)$
6:    **Assignment:** $\pi^{(t)} \leftarrow \arg\min_{\pi \in S_n} \mathrm{Trace}\big(\boldsymbol{C}_\pi^{(t)}\big)$    *(exact LSAP: Linear Sum Assignment Problem)*
7:    **Momentum update:** $\widehat{\boldsymbol{Z}}_i^{(t)} \leftarrow \rho\, \boldsymbol{Q}^n_{\pi^{(t)}(i)} + (1 - \rho)\, \widehat{\boldsymbol{Z}}_i^{(t-1)}$
8: **end for**
9: **Output:** $\widehat{\boldsymbol{\theta}} = \boldsymbol{\theta}^{(T)}$, aligned latents $\widehat{\boldsymbol{Z}} = \widehat{\boldsymbol{Z}}^{(T)}$.

---

### 3.5 COMPLEXITY AND SCALABILITY

Each outer iteration forms $C^{(t)}$ with $n$ forward passes over $\{z_k\}$ and solves one Hungarian problem: overall $O(n\, c_G + n^3)$, where $c_G$ is the decoder's forward cost. Crucially, the *assignment cost does not depend on $p$*, which is why NEUROSQL scales favorably to (very) high dimensions.

The $O(n^3)$ Hungarian algorithm may be an overhead for large sample sizes. To mitigate the cubic complexity for larger batches (in practice, for $n > 10^4$), we implemented and benchmarked a simpler yet robust greedy algorithm that is faster than the Hungarian algorithm and has complexity $O(n^2)$. More details on the greedy algorithm can be seen in section G. In our benchmarks, the greedy strategy yielded a $6.54\times$ speedup (reducing the mean execution time from $\approx$ 89ms to $\approx$ 13ms per call) while loading the full batch with no significant change in performance.

The change in method has no serious impact on the qualitative performance of the generated images; hence, the greedy algorithm could be used for use cases and domains with higher data demand. For data with ultra-large sample sizes that is challenging for full-batch assignment, one can consider NEUROSQL in a mini-batch setting, with a batch size $m \ll n$. By doing so, one can further reduce the complexity to $O(m^2)$, making the computational cost independent of the total dataset size $n$. A more precise ablation could be seen under G in the Appendix section.

### 3.6 INTERPRETABILITY OF THE LATTICE CODES

With NeuroSQL, our goal is to replace encoders in a stochastic manner with a deterministic, explainable assignment algorithm. Such an algorithm would yield a more structural, interpretable latent space for sampling.

To assess the interpretability of NEUROSQL in the latent space, we visualize its learned embeddings on the MNIST dataset in both 2D and 3D, and compare them against those obtained from a standard VAE baseline with the same generator architecture. Fig. 2 shows that the VAE latent space (Fig. 2b) exhibits the characteristic "fuzzy cloud" of the Gaussian prior $\mathcal{N}(0, I)$. This is likely because, while the VAE forces the aggregate posterior to approximate the prior via the KL divergence, the individual class manifolds (represented by color) exhibit significant overlap. This stochasticity obscures the decision boundaries between classes, making precise manipulation of the latent variables more difficult. In contrast, our NEUROSQL model (Fig. 2a) reveals a cleaner geometry. Because NEUROSQL approximates the latent variables by solving a linear assignment problem against a fixed grid, the resulting space is fully utilized and comparatively more regular. In the 2D projection (Fig. 2a), we

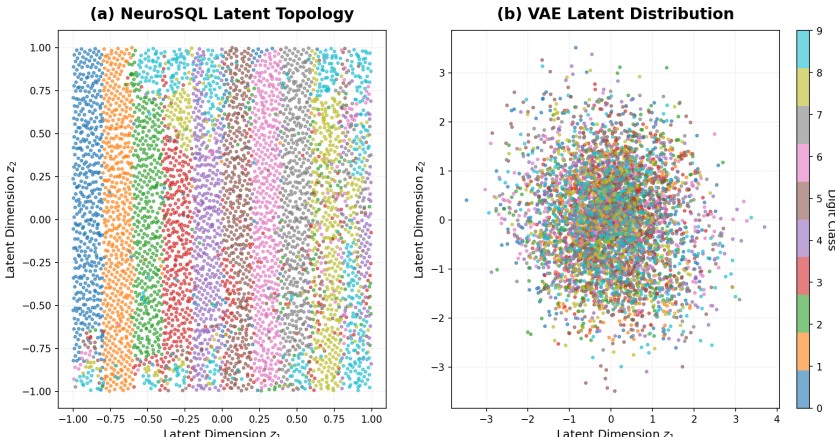

Figure 2: **A Comparison of Latent Space of NEUROSQL and VAE.** Visualization of the two-dimensional latent space obtained from the MNIST data using NEUROSQL (left) and VAE (right). While the VAE latent space appears diffused, the NEUROSQL latent space exhibits clear, stratified patterns corresponding to the ten digits.

observe a stratification of digit classes into vertical bands. This indicates that the optimal transport map has disentangled semantic class information along specific spatial dimensions without supervision. Our explorations suggest that NEUROSQL avoids the mode collapse often observed in GANs and the posterior collapse of VAE, providing a transparent, one-to-one mapping between the latent code and the generated sample.

## 4 DATA AND EXPERIMENTAL SETUP

We evaluate NEUROSQL under a *sparse-resource* regime across four data domains of increasing structural complexity: digits (MNIST), faces (CelebA), animal faces (AFHQ), and neuroimaging (OASIS). More details on the data can be found in section C. Our emphasis is on *paradigm evaluation*: models share (as closely as possible) the same generator backbone, data budgets, and optimization schedules, so that differences arise from the learning principle (quantile–assignment vs. probabilistic/adversarial/denoising) rather than model capacity or compute power.

### 4.1 MODELS AND TRAINING

We compare NEUROSQL to VAE, GAN, and DDPM under matched budgets and identical generator backbones; only the learning paradigm differs. For NEUROSQL we used the Sobol or Uniform lattice in $[0, 1]^d$ to construct the quantiles. We performed the *Hungarian* or *Greedy* algorithm at every $K$ epochs (with $K \in \{2, 3, 5\}$ treated as a hyperparameter), and selected the momentum parameter as $\rho = 0.7$. We trained NEUROSQL with the perceptual loss $\ell = \frac{1}{2}(1 - \text{SSIM}) + \frac{1}{2}\|\cdot\|_1$. All methods use AdamW, cosine decay, gradient clipping, and early stopping; full architectures and hyperparameters are in Appendix K.

**Generator backbones.** Unless otherwise specified, NeuroSQL/VAE/GAN share an identical *ConvNet upsampling decoder* (four residual up blocks, $512 \rightarrow 256 \rightarrow 128 \rightarrow 64$, $3 \times 3$ head with sigmoid). We ablate *Res*Net and *U-Net* decoders to show that gains are *decoder-agnostic*. For fairness, we chose diffusion's U-Net width so the parameter count is within $\pm 10\%$ of the shared decoder.

### 4.2 FAIRNESS CONTROLS AND ABLATIONS

**Backbone parity:** NeuroSQL/VAE/GAN share identical decoder weights at init; diffusion U-Net is matched by trainable parameter count. **Budget parity:** Same epochs, optimizer/scheduler, grad clipping, and batch size per dataset. **Latent sweep:** $q \in \{2, \ldots, 128\}$. **Resolution:** OASIS at $64^2$ vs. $128^2$. **Loss sensitivity:** pixel $\ell_2$ vs. SSIM+L1 for the assignment cost (main uses SSIM+L1; see details in the Appendix).

## 5  RESULTS

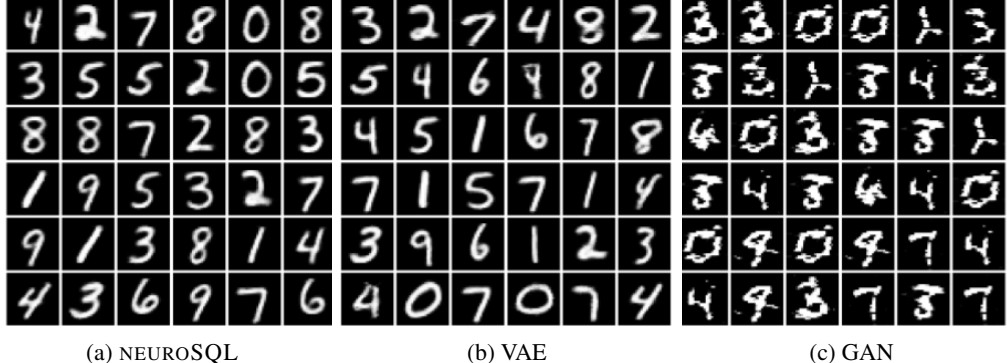

|  (a) NEUROSQL | (b) VAE | (c) GAN |

Figure 3: Qualitative comparison of 36 randomly generated images for the models trained on MNIST data.

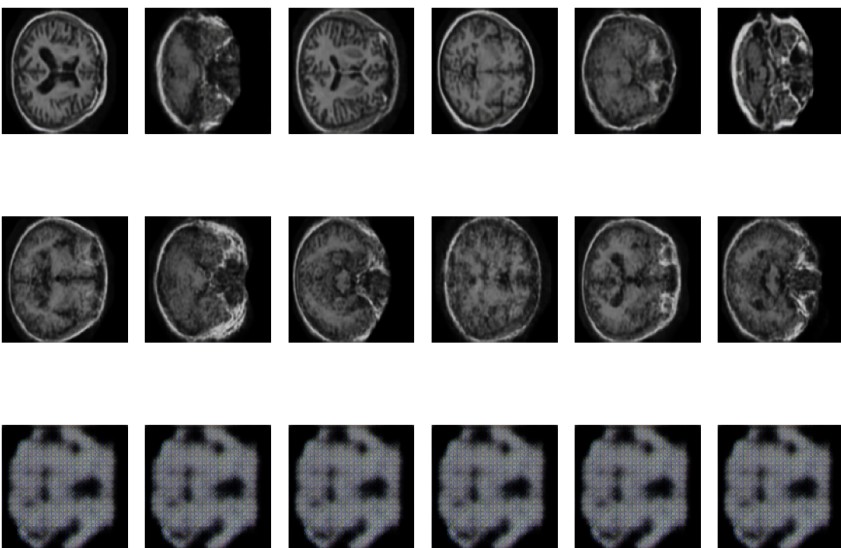

Figure 4: 2D Brain Images Generated from NEUROSQL (top row), VAE (middle row) and GAN (bottom row) with U-Net Generator.

### 5.1  OBSERVATIONS

We evaluate NeuroSQL's synthetic data generation against strong baselines across benchmarks. To ensure a fair comparison, we (1) test various latent dimensionalities, (2) include both ConvNet, ResNet, and U-Net generators, (3) evaluate all models on generating different types of data, and (4) we train a downstream classifier using synthetic images. For demonstration purposes, we discuss NEUROSQL 's performance against VAE and GAN on generating brain imaging data and MNIST digits, but see the Appendix for comparison with diffusion models and model performance on human and animal faces. To quantify the model performance, we report, for each scenario, a proxy of FID (Fréchet Inception Distance; lower is better), LPIPS (Learned Perceptual Image Patch Similarity; lower is better), and SSIM (Structural Similarity Index; higher is better). Discussion on the metrics and datasets could be found in the Appendix under C and D. We present the results across a wide range of experimental specifications in several tables in Appendix N.

For the brain imaging data (OASIS), we observe that, overall, NEUROSQL outperforms both VAE and GAN across various combinations of latent dimension and generator. Particularly, NEUROSQL outperforms VAE and GAN in terms of LPIPS (measuring perceptual similarity between images

Table 1: Performance comparison of generative models across datasets under resource-constrained settings. Results reported as mean ± std across architectures and latent dimensions. Lower is better for FID; higher is better for SSIM.

| Dataset | Resolution | Model | proxy FID↓ | SSIM↑ | Params (M) |
|---------|-----------|-------|-----------|-------|-----------|
| **MNIST** | 28×28 | NEUROSQL | **0.61±0.12** | **0.616±0.074** | **2.79** |
| | | VAE | 1.26±0.26 | 0.179±0.030 | 4.17 |
| | | GAN | 2.00±0.22 | 0.233±0.018 | 5.56 |
| | | Diffusion | 0.65±0.34 | 0.492±0.055 | 147.91 |
| **CelebA** | 64×64 | NEUROSQL | **5.81±4.05** | **0.262±0.036** | **7.54** |
| | | VAE | 10.75±7.92 | 0.196±0.047 | 14.03 |
| | | GAN | 18.02±6.78 | 0.137±0.034 | 10.33 |
| | | Diffusion | 23.79±8.11 | 0.013±0.066 | 147.91 |
| **AFHQ** | 128×128 | NEUROSQL | **19.03±11.31** | **0.290±0.056** | **119.58** |
| | | VAE | 39.15±30.64 | 0.190±0.051 | 144.98 |
| | | GAN | 46.00±14.83 | 0.082±0.029 | 122.37 |
| | | Diffusion | 22.99±14.83 | 0.0388±0.032 | 147.91 |
| **OASIS** | 128×128 | NEUROSQL | **16.36±12.25** | **0.252±0.010** | **243.74** |
| | | VAE | 24.24±24.46 | 0.196±0.058 | 269.09 |
| | | GAN | 68.23±15.52 | 0.145±0.063 | 246.50 |
| | | Diffusion | 21.22±23.71 | 0.04759±0.058 | 252.66 |

(i.e., how similar they look to humans) and SSIM (measuring pixel-level structural similarity) scores in all scenarios. All models with a ConvNet generator yield much better results compared to those with a ResNet. The choice of generator, however, has a marginal effect on FID scores. For FID, which measures distribution similarity between generated and real images in feature space and pixel distance, NEUROSQL outperforms VAE and GAN in all cases with a ResNet generator. It outperforms VAEs and GANs when the latent dimension is moderate (between 16 and 64) with a ConvNet generator. When the latent dimension is very small or very large, VAE with ConvNet shows only modest improvement. With a U-Net generator, NEUROSQL attains the best LPIPS and SSIM, while the FID (proxy) varies and is lowest for VAE on average.

Taken together, our results demonstrate the overall efficacy of NEUROSQL compared to its competitors in generating synthetic data across different domains that are unrelated to each other. In particular, NEUROSQL with a ConvNet generator achieves superior performance across different scenarios, particularly in metrics evaluating perceptual similarity between images (how similar they look to humans) and pixel-level structural similarity, and its performance improves as the dimension of its latent space increases.

A more comprehensive quantitative comparison is detailed in Table 1, which reports the mean performance metrics across all tested architectures and latent dimensions. These results confirm the overall assessment that NEUROSQL delivers the most robust balance of image quality and structural similarity across digits, faces, and medical imaging data with the smallest amount of trainable parameters.

## 5.2 COMPUTE BUDGET, DATA SCALE, AND EVALUATION SCOPE

Our aim is to introduce a learning *paradigm*, not to chase unmatched image quality. All experiments ran end-to-end on a single Google Colab with a fixed allowance of *200 compute units*. Within this resource budget we capped training data at **2,000 images** and resolutions at 64×64–128×128. This budget dictated architectures, optimisers, and protocols, targeting *compute- and data-frugal* generative learning. These constraints place our work within the emerging area of *resource-constrained generative modeling*, where practical applications, including, although not limited to, medical imaging, rare species documentation, and scientific datasets, cannot leverage massive data or compute budgets typical of modern generative AI.

Given these computational constraints, we carefully select evaluation domains where low-resolution generation remains semantically meaningful. We visualize only benchmarks whose semantics sur-

vive low resolution: (i) OASIS 2D brain MRI slices, where gross anatomy is discernible at 64–128 px, and (ii) MNIST digits, an intrinsically low-frequency signal. For faces/natural scenes, perceptual judgments at 64–128 px are less diagnostic under our budget; raising resolution would, however, go beyond the cap. Thus, we highlight that our contribution is the *training principle*, not a new high-capacity image backbone.

Understanding which existing methods are suitable for our constrained pipeline is equally important. In particular, while diffusion models achieve state-of-the-art results at scale, they are not well-suited to this task. With $T=1000$ steps and $N\approx1000$ images, supervision per noise level is $O(N/T)$, yielding high-variance score estimates; a single noise-conditional network must span a wide range of Signal to Noise Ratio (SNR) under tight capacity and time, and gradients are diluted by averaging across many timesteps. Moreover, wall-clock scales with $T$, reducing optimizer updates at a fixed budget. Low-data remedies (heavy augmentation, distillation, large-scale pretraining) fall outside our *from-scratch, small-budget* premise. The observed poor performance of the DDPM baseline highlights the specific need for NEUROSQL to handle small- or moderate-$n$ and large-$p$ problems under limited computational resources.

To facilitate reproducibility and establish the broader applicability of our approach, we provide the following practical considerations. All settings are reproducible within the stated Colab budget. Future work should systematically compare NEUROSQL against other efficient generative methods designed for similar resource constraints (e.g., FastGAN (Liu et al., 2021), Projected GAN (Sauer et al., 2021)) to establish benchmarks within this paradigm.

## 6 CONCLUSIONS

We introduced NEUROSQL , a deep generative model that replaces stochastic encoders with rank-based quantile assignment. By learning embeddings through assignments rather than amortised inference, NEUROSQL eliminates the mode of posterior collapse in VAE and avoids adversarial dynamics, yielding more stable training than GANs. Thanks to the quantile assignment, NEUROSQL 's generation processes are deterministic, resulting in distributions that are substantially more interpretable than those of GANs and diffusion models, and more interpretable than VAEs, whose embeddings depend on samples from a Gaussian posterior, even when the encoder is deterministic. Compared to diffusion models, which require extensive denoising steps and large datasets for stable training, NEUROSQL demonstrates better performance in constrained settings with datasets under $10^5$ samples. The deterministic assignment mechanism avoids the high-variance score estimation that plagues diffusion models in low-data regimes. Trained on four distinctive domains: handwritten digits (MNIST), human faces (CelebA), animal faces (AFHQ), and brain images (OASIS) to generate synthetic samples, NEUROSQL is consistently competitive and often superior, with the best performance under the same experimental settings.

Finally, we emphasize that our main contribution is the quantile-assignment training principle itself, rather than a new high-capacity architecture for large-scale image generation. While our constrained experimental setup was intentional to introduce the NEUROSQL paradigm under controlled conditions, future work should address its scalability to larger datasets and higher resolutions, compare it with advanced baselines like StyleGAN3 and latent diffusion models, and explore alternatives to the $O(n^3)$ Hungarian algorithm, such as approximate Sinkhorn iterations. In terms of future applications, for example, that of medical data science, a natural next step is to experiment with the MedMNIST (Yang et al., 2023) benchmark suite to assess NEUROSQL across diverse biomedical imaging tasks. Moving forward, we see two promising directions: (i) scaling the assignment mechanism and generator capacity to higher resolutions and additional modalities (e.g., audio, 3D, and multimodal settings), and (ii) expanding the theory of quantile-assignment training for deep generators beyond the current empirical scope.

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

## A    ACKNOWLEDGMENTS

We acknowledge the use of large language models for grammar checking, punctuation correction, spelling verification, and synonym suggestions to enhance writing clarity.

## B    ETHICAL AND DOMAIN-SPECIFIC NOTES (OASIS)

In order to keep the experimental setup free of data leakage within the train/val/test splits, we performed subject-stratified cross-validation. We highlight that one must not equate synthetic imaging data with clinical data. Synthetic imaging data, however, may have practical utilities, such as for treating missing data, but this is beyond the scope of this paper. Here, we generate synthetic data to demonstrate the efficacy of NEUROSQL as a generative model; we use the generated imaging data to evaluate the model's performance; we do not claim its clinical utility. Future work should verify this independently, and we will release seeds, splits, and scripts to facilitate further validations.

## C    DATASETS

**MNIST (LeCun et al., 2002).**    Handwritten digits (60k train, 10k test, 28×28 grayscale). We replicate to RGB for evaluation only.

**CelebA (Liu et al., 2015).**    Face dataset (202,599 images) center-cropped and resized to 128×128. Training is unconditional despite available attributes.

**AFHQ (Choi et al., 2020).**    Animal Faces HQ contains 15,000 high-quality animal face images across cats, dogs, and wildlife at 512×512 resolution originally. Within experiments, image size was reduced to 128×128 and total number of images used was about 2000.

**OASIS (Marcus et al., 2007; Aithal, 2023).**    Neuroimaging dataset with $\sim$ 80k MRI slices from 416 subjects, downsampled to 128×128. Medical images test resistance to overfitting in constrained domains. We used the version of OASIS that is publicly available on Kaggle. It is pre-structured and is organized in four subfolders, namely: Mild Dementia, Moderate Dementia, Non Demented, and Very Mild Demented.

## D    EVALUATION METRICS

**FID (proxy; ↓):** Since our domains are not ImageNet/COCO, we report FID only as a coarse proxy (Inception-V3 pool3, 2048-d); in this setting, it behaves like a *lower mean pixel distance* between synthetic and real sets. We match #gen=#real (MNIST: 10k; CelebA: test split; OASIS: test slices), fix the sampling seed, and replicate grayscale to 3 channels at metric time.    **LPIPS (↓):** VGG backbone via LPIPS (fallback: cosine similarity on VGG features); mean over 50 paired real–fake images.    **SSIM (↑):** mean over the same 50 pairs.    Unless noted, we evaluate NEUROSQL in *sampled-z* mode ($z \sim \mathcal{N}(0, I)$); *paired-z* (reconstruction) results appear in the supplement.

## E    PREPROCESSING AND SPLITS

For each dataset, we standardize a lightweight pipeline to minimize confounds while allowing small variations across runs.

- **Resize/crop.** MNIST: native 28×28. CelebA: center-crop then resize, typically to 32×32 (primary), with occasional 64–128 experiments. OASIS: center-crop then resize, primarily 128×128 with 64×64 ablations.
- **Scaling.** Inputs mapped to $[0, 1]$ (no per-image standardization during training).
- **Channel handling.** MNIST/OASIS trained as single-channel; for backends expecting 3 channels (e.g., LPIPS/VGG), we replicate channels *at metric time only*.
- **Splits.** MNIST: standard train/test. CelebA: official train/val/test. OASIS: subject-wise 80/10/10 to prevent slice leakage. Seeds and split indices are provided in the supplementary.

**Scope and reproducibility.** Settings are chosen for a small-resource envelope (single-session runs). Results emphasize methodology under constrained compute; scaling to larger images/batches follows the same code paths.

**Optimization and budgets (typicals/ranges).** AdamW (or Adam), cosine warm restarts; weight decay $\sim 10^{-4}$; gradient clip $\approx 1.0$; early stopping on validation loss with patience $\sim$ 25–30 epochs. Learning rates are usually in $[1 \times 10^{-4}, 3 \times 10^{-4}]$ for convolutional decoders; diffusion runs use comparable schedules at matched compute. Batch sizes depend on resolution: `MNIST` 32–64, `CelebA` 64–128, `OASIS` 32–64. Epoch caps are typically 120–250 across datasets, with early stopping often terminating earlier. We sweep latent dimension $q$ over $\{2, 4, 8, 16, 32, 64, 128\}$ and report results (Sec. N).

## F  MINI-BATCH TRAINING NEUROSQL

While Algorithm 1 describes the exact full-batch procedure, NEUROSQL scales to large datasets ($n \gg 10^4$) via a stochastic approximation outlined in Algorithm 2. In this regime, we maintain a persistent "memory bank" of latent codes $\widehat{Z}$. In each iteration, we sample a mini-batch of data $X_{\mathcal{B}}$ and a corresponding random subset of the lattice $Q_{\mathcal{K}}$. The assignment problem is solved locally on the $m \times m$ cost matrix. This reduces the computational complexity of the assignment step from $O(n^3)$ to $O((n/m) \cdot m^3) = O(n \cdot m^2)$ per epoch using the Hungarian method, or $O(n \cdot m^2)$ using the Greedy alternative. The momentum parameter $\rho$ is critical here, acting as a temporal smoother that stabilizes the stochastic assignment trajectory towards the optimal transport map.

---

**Algorithm 2** Quantile Assignment in Mini-batch setting (Mini-batch / Large Scale)

---

1: **Input:** data $\{X_i\}_{i=1}^n$, prior $P_Z$, lattice $Q^n$, batch size $m$, epochs $E$, momentum $\rho \in [0, 1)$
2: **Initialize:** Global latent codes $\widehat{Z} = \{\widehat{Z}_1, \ldots, \widehat{Z}_n\}$ initialized via random $Q^n$ assignment.
3: **for** epoch $e = 1, \ldots, E$ **do**
4:     Shuffle data indices $\{1, \ldots, n\}$
5:     **for** batch indices $\mathcal{B} \subset \{1, \ldots, n\}$ with $|\mathcal{B}| = m$ **do**
6:         **1. Generator Update:**
7:
$$\boldsymbol{\theta} - \eta \nabla_{\boldsymbol{\theta}} \left[ \frac{1}{m} \sum_{i \in \mathcal{B}} \ell\big(X_i, G_{\boldsymbol{\theta}}(\widehat{Z}_i)\big) \right]$$

8:         **2. Stochastic Quantile Assignment:**
9:         Sample lattice subset indices $\mathcal{K} \subset \{1, \ldots, n\}$ of size $m$.
10:         Compute batch cost matrix $C \in \mathbb{R}^{m \times m}$:
11:             $C_{j,k} \leftarrow \ell\big(X_{\mathcal{B}[j]}, G_{\boldsymbol{\theta}}(Q_{\mathcal{K}[k]})\big)$ for $j, k \in \{1 \ldots m\}$
12:         Solve assignment $\pi_{\text{batch}}$ on $C$ (via Hungarian or Greedy).
13:         **3. Momentum Update (Sparse):**
14:         For each $j \in \{1 \ldots m\}$, update global memory bank:
15:             $\widehat{Z}_{\mathcal{B}[j]} \leftarrow \rho \, Q_{\mathcal{K}[\pi_{\text{batch}}(j)]} + (1 - \rho) \, \widehat{Z}_{\mathcal{B}[j]}$
16:     **end for**
17: **end for**
18: **Output:** Generator $G_{\boldsymbol{\theta}}$, aligned latents $\widehat{Z}$.

---

## G  GREEDY ASSIGNMENT ALGORITHM

To address the scalability limitations of the Hungarian algorithm (which scales as $\mathcal{O}(n^3)$) for larger batch sizes or datasets, we implement a Greedy Assignment strategy. While the Hungarian algorithm guarantees the global minimum cost for the linear assignment problem, the Greedy approach provides an approximation that is computationally efficient ($\mathcal{O}(n^2)$) and sufficient for maintaining training stability in the paradigm we introduce with our model.

**Algorithm Description.** The greedy strategy iterates over each row of the cost matrix. For each row $i$, it selects the column $j$ that minimizes the cost $C_{i,j}$, provided that column $j$ has not already

been assigned to a previous row. Once a column is selected, it is removed from the pool of available columns.

---

**Algorithm 3** Greedy Assignment ($\mathcal{O}(n^2)$)

---

1: **Input:** Cost matrix $C \in \mathbb{R}^{n \times n}$
2: **Initialize:** Set of assigned columns $\mathcal{S} \leftarrow \emptyset$, Permutation vector $\pi$ of size $n$
3: **for** $i = 0$ **to** $n - 1$ **do**
4:     $min\_cost \leftarrow \infty$
5:     $best\_col \leftarrow -1$
6:     **for** $j = 0$ **to** $n - 1$ **do**
7:         **if** $j \notin \mathcal{S}$ **and** $C_{i,j} < min\_cost$ **then**
8:             $min\_cost \leftarrow C_{i,j}$
9:             $best\_col \leftarrow j$
10:         **end if**
11:     **end for**
12:     $\pi[i] \leftarrow best\_col$
13:     $\mathcal{S} \leftarrow \mathcal{S} \cup \{best\_col\}$
14: **end for**
15: **Return:** Row indices $\{0, \ldots, n - 1\}$, Column indices $\pi$

---

**Computational Complexity.** The outer loop runs exactly $n$ times (once for each data sample). The inner loop scans $n$ columns to find the minimum unassigned cost. Although the number of available columns decreases by 1 in each iteration, the upper bound of the search remains $n$. Consequently, the total complexity is proportional to $\sum_{i=1}^{n} n = n^2$, yielding a time complexity of $\mathcal{O}(n^2)$.

With Table 2 we try to show the quantitative difference that occurs when the assignment algorithm changes.

Table 2: Ablation Study: U-Net Latent Dimensions

| Latent Dim | Method | FID (proxy) ↓ | LPIPS ↓ | SSIM ↑ |
|---|---|---|---|---|
| 8 | SQL-greedy | 10.764394 | 0.376178 | 0.296806 |
| 8 | SQL-hungarian | 11.121147 | 0.378670 | 0.274640 |
| 16 | SQL-greedy | 11.632382 | 0.398841 | 0.241317 |
| 16 | SQL-hungarian | 11.798410 | 0.390295 | 0.240504 |
| 24 | SQL-greedy | 13.222514 | 0.384961 | 0.272349 |
| 24 | SQL-hungarian | 11.338643 | 0.379329 | 0.271508 |

## H MNIST DOWNSTREAM CLASSIFIER

To be sure of the quality of the generated samples from NEUROSQL we decided to train a downstream classifier using sampled images. We trained a standard CNN classifier solely on synthetic images generated by our model (15k) and evaluated it on real MNIST data. The model achieved **90.27% accuracy, 0.90 precision, and 0.90 recall**. Therefore, we believe that the generated samples effectively capture the class-conditional distributions of the underlying manifold.

## I GENERATED IMAGES

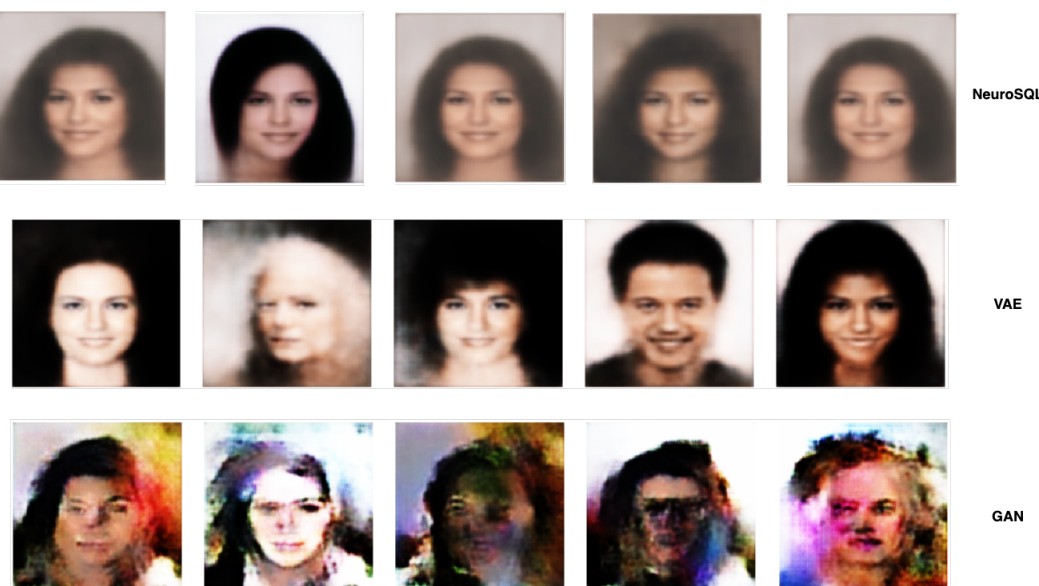

Figure 5: Qualitative comparison of NeuroSQL against standard generative models across different decoder architectures. Generated samples from CelebA faces

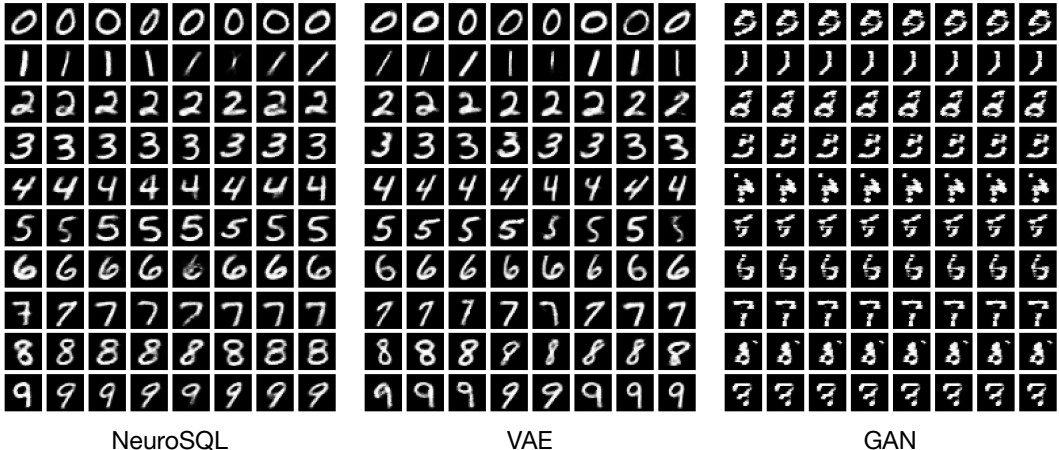

Figure 6: **MNIST digit generation comparison across generative modeling paradigms.** Each method generates 8 samples per digit class (0–9) using identical network capacities and training budgets. NeuroSQL (left) produces consistently sharp, well-formed digits with clear class separation and minimal artifacts. VAE (center) exhibits typical reconstruction blur and shape distortions, particularly evident in digits with fine details (e.g., 8, 9). GAN (right) shows characteristic training instabilities, including mode collapse, with several digit classes producing nearly identical or malformed samples. The systematic comparison demonstrates NeuroSQL's ability to maintain both sample quality and diversity across all digit classes, highlighting the advantages of the deterministic quantile-assignment approach over stochastic generative methods in controlled generation tasks.

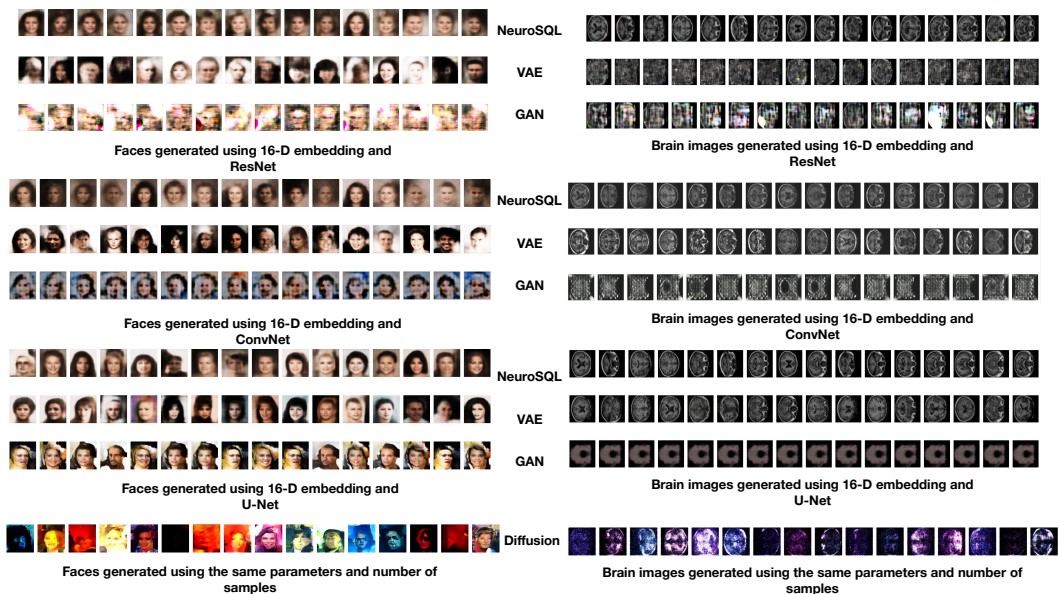

Figure 7: **Qualitative comparison of NeuroSQL against standard generative models across different decoder architectures.** Generated samples from CelebA faces (left) and brain MRI images (right) using 16-dimensional latent embeddings. Each row shows results from a different generative approach: NeuroSQL (ours), VAE, GAN, and Diffusion (DDPM). For NeuroSQL, we demonstrate consistency across three lightweight decoder architectures (ResNet, ConvNet, U-Net), highlighting that performance gains stem from the quantile-assignment paradigm rather than decoder sophistication. VAE and GAN results show typical artifacts, including mode collapse, blurriness, and training instabilities, while diffusion models exhibit characteristic oversaturation and unrealistic color distributions. NeuroSQL produces consistently high-quality, diverse samples across all decoder choices, demonstrating the robustness of the deterministic lattice-based approach. All models use identical training budgets and matched generator capacities for fair comparison.

## J   QUANTITATIVE EVALUATION OF RUNTIME AND MEMORY

Here, we would like to highlight and provide a quantitative evaluation of the memory and runtime of our proposed model. The following Table 3 is based on the metrics gathered from an MNIST run.

Table 3: Ablation: Assignment Methods and Resource Usage on MNIST (seed=11). Metrics averaged over epochs where applicable.

| Latent Dim | Assign. Method | Method | Peak VRAM (MB) | RAM Used (MB) | Mean Epoch Time (s) | Mean Epoch GPU (MB) | Mean Epoch ΔRAM | FID (proxy)↓ | LPIPS↓ | SSIM↑ |
|---|---|---|---|---|---|---|---|---|---|---|
| 2 | greedy hungarian | NeuroSQL | 81.5 | 2764 | 1.40 | 116 | 11.2 | 0.628 | 0.045 | 0.580 |
| | | NeuroSQL | 284.6 | 3078 | 1.37 | 293 | 0.39 | 0.576 | 0.041 | 0.616 |
| | | VAE | 319.8 | 3091 | 1.33 | 320 | 0.10 | 1.259 | 0.050 | 0.249 |
| | | GAN | 405.6 | 3115 | 1.56 | 406 | 0.17 | 2.060 | 0.064 | 0.223 |
| 3 | greedy hungarian | NeuroSQL | 284.8 | 3206 | 1.39 | 293 | 0.14 | 0.567 | 0.040 | 0.574 |
| | | NeuroSQL | 284.9 | 3298 | 1.38 | 293 | 0.00 | 0.556 | 0.037 | 0.607 |
| | | VAE | 320.7 | 3284 | 1.34 | 321 | 0.10 | 1.544 | 0.053 | 0.185 |
| | | GAN | 404.6 | 3293 | 1.56 | 404 | 0.01 | 5.443 | 0.327 | 0.070 |

## K  MODELS AND TRAINING PROTOCOL

**Common setup.**  Images are scaled to $[0, 1]$ (diffusion uses $[-1, 1]$ internally). We use AdamW, cosine annealing, gradient clipping, and early stopping on validation loss. Generator backbones are matched across methods for capacity parity.

**NeuroSQL (ours).**  We construct a size-$n$ deterministic latent lattice via scrambled Sobol points mapped coordinatewise through $F_{Z_\ell}^{-1}$ (Sobol→Gaussian). Every $K$ epochs we solve an exact global assignment (Hungarian) between data and lattice codes, where $K \in \{2, 3, 5\}$ is selected as a hyper-parameter. After each assignment, we apply latent momentum $\widehat{z}^{(t)} \leftarrow \rho\, z_{\pi^{(t)}(i)} + (1 - \rho)\, \widehat{z}^{(t-1)}$ with $\rho = 0.7$. The decoder is trained by regression on assigned codes using $\ell = \frac{1}{2}(1 - \text{SSIM}) + \frac{1}{2}\|\cdot\|_1$.

**VAE.**  We reuse the same generator backbone as the decoder; the encoder is an MLP on flattened pixels (to keep capacity modest). Training uses SSIM+L1 reconstruction plus a $\beta$-scaled KL term with $\beta = 0.005$.

**GAN.**  The generator backbone is identical to NeuroSQL's. The discriminator is a lightweight four-layer CNN. We use the non-saturating objective with BCE logits, sharing optimiser, scheduler, and gradient clipping with NeuroSQL.

**Diffusion (DDPM).**  A compact U-Net (base width 32) is trained with a linear $\beta$ schedule for $T = 1000$ steps; default sampling uses 100 steps to match compute. Inputs are normalized to $[-1, 1]$ following common practice.

**Reproducibility knobs.**  We fix random seeds, match the number of training epochs and batch sizes across methods, and report all per-method hyperparameters (including learning rates, $K \in \{2, 3, 5\}$, and augmentations).

## L  PROOF OF PROPOSITION 1

*Proof.* Following Hallin et al. (2021), we define the center-outward empirical distribution function as the solution of the optimal transportation problem:

$$F_\pm^n = \arg\min_{T \in \mathcal{T}} \sum_{i=1}^n \|\boldsymbol{Z}_i - T(\boldsymbol{Z}_i)\|^2,$$

where the minimum is taken over $\mathcal{T}$, the set of all bijective mappings between $\boldsymbol{Z}_1, ..., \boldsymbol{Z}_n$ and the grid $\boldsymbol{U}_1, \ldots, \boldsymbol{U}_n$. In fact, this is equivalent to solve a linear assignment problem:

$$\pi^* = \arg\min_{\pi \in S_n} \|\boldsymbol{Z} - \boldsymbol{U}_\pi^n\|^2,$$

and set $F_\pm^n(\boldsymbol{Z}^n) := \boldsymbol{U}_{\pi^*}^n$. Then we apply Theorem 2.4 in Hallin et al. (2021) to obtain:

$$\max_{1 \le i \le n} \|F_\pm^n(\boldsymbol{Z}_i) - F_\pm(\boldsymbol{Z}_i)\|^2 \to 0 \text{ a.s. as } n \to \infty.$$

In our case, as $F_\pm$ is the uniform distribution over $\mathcal{U}_d$, it is relatively straightforward to see that:

$$\min_{\pi \in S_n} \|\boldsymbol{Q}_\pi^n - \boldsymbol{Z}\|^2 \to 0, \text{ a.s. as } n \to \infty.$$

$\square$

## M  PRACTICAL TRAINING DETAILS

**Loss choices.**  For images, we use a perceptual, scale-stable loss:

$$\ell(\hat{x}, x) = \frac{1}{2}\big(1 - \text{SSIM}(\hat{x}, x)\big) + \frac{1}{2}\|\hat{x} - x\|_1,$$

which is the exact loss used in our codebase.

We instantiate gen with lightweight decoders so that comparisons against VAEs/GANs/Diffusion control for capacity and compute:

- **ConvNet.** Transposed-convolution stack mapping $z \in \mathbb{R}^q$ to $X \in \mathbb{R}^{3 \times H \times H}$ (stride-2 upsampling).

- **ResNet** Four residual upsampling blocks (512→256→128→64), followed by a $3 \times 3$ head with sigmoid output in $[0, 1]$. We optionally initialize residual weights from ResNet-18 where shapes match.

- **U-Net decoder.** A small transformer decoder on patchified embeddings of $z$ followed by an MLP head back to pixels.

Our experiments keep these decoders small and matched across methods to stress that gains come from the *quantile–assignment loop*, not decoder sophistication.

- **Loss and normalization.** Images are scaled to $[0, 1]$. We use $\ell = \frac{1}{2}(1 - \mathrm{SSIM}) + \frac{1}{2}\ell_1$ in both decoder and cost matrix.

- **Optimization.** AdamW with cosine annealing and gradient clipping; early stopping on validation $\ell$.

- **Latent momentum.** After each assignment, a momentum update $\widehat{z}^{(t)} \leftarrow \rho\, z_{\pi^{(t)}(i)} + (1 - \rho)\, \widehat{z}^{(t-1)}$ stabilizes training (we use $\rho = 0.7$).

- **Resource parity.** For fair comparisons to VAEs, GANs, and Diffusion, we fix the *same* generator backbone and training budget; only the learning paradigm changes.

## N   TABULAR RESULTS ON OASIS, CELEBA, AFHQ AND MNIST

Table 4: Results on OASIS, a brain imaging dataset, using ConvNet across latent dimensions. Lower is better for FID (proxy) and LPIPS; higher is better for SSIM.

| Latent dimension | Method | FID (Proxy) ↓ | LPIPS ↓ | SSIM ↑ |
|---|---|---|---|---|
| | NeuroSQL | 7.914 | 0.390 | 0.259 |
| 2 | VAE | 8.198 | 0.410 | 0.246 |
| | GAN | 9.936 | 0.450 | 0.218 |
| | NeuroSQL | 9.241 | 0.403 | 0.257 |
| 4 | VAE | 8.720 | 0.463 | 0.212 |
| | GAN | 19.420 | 0.509 | 0.206 |
| | NeuroSQL | 8.286 | 0.411 | 0.256 |
| 8 | VAE | 8.029 | 0.485 | 0.215 |
| | GAN | 12.766 | 0.558 | 0.193 |
| | NeuroSQL | 8.602 | 0.455 | 0.267 |
| 16 | VAE | 12.768 | 0.539 | 0.171 |
| | GAN | 12.485 | 0.584 | 0.200 |
| | NeuroSQL | 8.856 | 0.401 | 0.257 |
| 32 | VAE | 12.490 | 0.525 | 0.178 |
| | GAN | 14.852 | 0.593 | 0.174 |
| | NeuroSQL | 7.385 | 0.388 | 0.265 |
| 64 | VAE | 16.003 | 0.567 | 0.151 |
| | GAN | 14.453 | 0.602 | 0.125 |
| | NeuroSQL | 18.743 | 0.453 | 0.248 |
| 128 | VAE | 16.329 | 0.571 | 0.158 |
| | GAN | 29.792 | 0.620 | 0.084 |

Table 5: Results on OASIS, a brain imaging dataset, using ResNet across latent dimensions. Lower is better for FID (proxy) and LPIPS; higher is better for SSIM.

| Latent dimension | Method | FID (proxy) ↓ | LPIPS ↓ | SSIM |
|---|---|---|---|---|
| | NeuroSQL | 25.410 | 0.249 | 0.301 |
| 16 | VAE | 51.139 | 0.309 | 0.171 |
| | GAN | 168.993 | 0.727 | 0.008 |
| | NeuroSQL | 31.957 | 0.257 | 0.242 |
| 32 | VAE | 66.045 | 0.362 | 0.128 |
| | GAN | 158.103 | 0.687 | 0.008 |
| | NeuroSQL | 30.892 | 0.220 | 0.221 |
| 64 | VAE | 47.146 | 0.358 | 0.135 |
| | GAN | 156.088 | 0.741 | 0.135 |
| | NeuroSQL | 34.346 | 0.224 | 0.196 |
| 128 | VAE | 52.317 | 0.397 | 0.135 |
| | GAN | 156.288 | 0.723 | 0.124 |

Table 6: Results on OASIS, a brain imaging dataset, using a U-Net across latent dimensions. Lower is better for FID (proxy) and LPIPS; higher is better for SSIM.

| Latent dimension | Method | FID (proxy) ↓ | LPIPS ↓ | SSIM ↑ |
|---|---|---|---|---|
| 4 | NeuroSQL | 8.519902 | 0.386861 | 0.254752 |
| 4 | VAE | 7.858056 | 0.377805 | 0.265082 |
| 4 | GAN | 26.234322 | 0.597633 | 0.206676 |
| 8 | NeuroSQL | 8.579974 | 0.401890 | 0.238080 |
| 8 | VAE | 5.976874 | 0.384821 | 0.264277 |
| 8 | GAN | 35.444614 | 0.609803 | 0.172225 |
| 16 | NeuroSQL | 10.037155 | 0.391049 | 0.260999 |
| 16 | VAE | 7.178138 | 0.402099 | 0.231028 |
| 16 | GAN | 20.907921 | 0.616312 | 0.225444 |
| 32 | NeuroSQL | 8.405630 | 0.388190 | 0.253740 |
| 32 | VAE | 4.424680 | 0.402910 | 0.238710 |
| 32 | GAN | 28.069950 | 0.613550 | 0.200620 |
| 64 | NeuroSQL | 8.259449 | 0.385646 | 0.262159 |
| 64 | VAE | 6.098939 | 0.395427 | 0.272117 |
| 64 | GAN | 30.385052 | 0.644507 | 0.153054 |
| 128 | NeuroSQL | 7.553965 | 0.371082 | 0.282864 |
| 128 | VAE | 9.101107 | 0.415775 | 0.257312 |
| 128 | GAN | 30.505713 | 0.553681 | 0.202557 |

Table 7: Results on OASIS, a brain imaging dataset, using a U-Net — with results averaged across latent dimensions $\{4, 8, 16, 32, 64, 128\}$.

| Method | FID (proxy) ↓ | LPIPS ↓ | SSIM ↑ |
|---|---|---|---|
| NeuroSQL | 8.559346 | **0.387453** | **0.258766** |
| VAE | **6.772966** | 0.396473 | 0.254754 |
| GAN | 28.591262 | 0.605914 | 0.193429 |

On average, across latent dimensions, NeuroSQL attains the best LPIPS and SSIM, while VAE has the lowest FID (proxy).

Table 8: Results on CelebA, a face attributes dataset, using ConvNet across latent dimensions. Lower is better for FID (proxy) and LPIPS; higher is better for SSIM.

| Latent dimension | Method | FID (proxy) ↓ | LPIPS ↓ | SSIM ↑ |
|---|---|---|---|---|
| | NeuroSQL | 9.64453 | 0.22094 | 0.31376 |
| 2 | VAE | 11.49087 | 0.24068 | 0.27396 |
| | GAN | 26.56548 | 0.47277 | 0.10216 |
| | NeuroSQL | 6.99789 | 0.22845 | 0.31645 |
| 4 | VAE | 8.57467 | 0.23091 | 0.26471 |
| | GAN | 29.68144 | 0.49874 | 0.06955 |
| | NeuroSQL | 6.686607 | 0.244427 | 0.297947 |
| 8 | VAE | 32.183292 | 0.281024 | 0.226046 |
| | GAN | 28.772638 | 0.461147 | 0.058802 |
| | NeuroSQL | 17.252127 | 0.304758 | 0.296877 |
| 16 | VAE | 11.955338 | 0.282781 | 0.160219 |
| | GAN | 20.922581 | 0.419844 | 0.117695 |
| | NeuroSQL | 4.04047 | 0.19269 | 0.25707 |
| 32 | VAE | 6.57785 | 0.21120 | 0.13975 |
| | GAN | 13.92531 | 0.31649 | 0.12201 |
| | NeuroSQL | 3.94787 | 0.20649 | 0.25782 |
| 64 | VAE | 3.93301 | 0.21478 | 0.15485 |
| | GAN | 16.27481 | 0.35050 | 0.10456 |
| | NeuroSQL | 4.91280 | 0.20557 | 0.26119 |
| 128 | VAE | 5.57425 | 0.20361 | 0.16837 |
| | GAN | 17.16048 | 0.34640 | 0.15798 |

Table 9: Results on CelebA, a face attributes dataset, using ResNet across latent dimensions. Lower is better for FID (proxy) and LPIPS; higher is better for SSIM.

| Latent dimension | Method | FID (proxy) ↓ | LPIPS ↓ | SSIM ↑ |
|---|---|---|---|---|
| | NeuroSQL | 4.40599 | 0.18367 | 0.27506 |
| 2 | VAE | 8.01552 | 0.19720 | 0.24972 |
| | GAN | 19.01699 | 0.20240 | 0.13535 |
| | NeuroSQL | 4.47511 | 0.16968 | 0.26754 |
| 4 | VAE | 6.03891 | 0.20652 | 0.20760 |
| | GAN | 14.21587 | 0.26898 | 0.14515 |
| | NeuroSQL | 4.60623 | 0.18201 | 0.25536 |
| 8 | VAE | 4.86620 | 0.25218 | 0.17823 |
| | GAN | 13.94172 | 0.18593 | 0.17628 |
| | NeuroSQL | 3.99240 | 0.18115 | 0.21823 |
| 32 | VAE | 6.00124 | 0.25017 | 0.12915 |
| | GAN | 11.08812 | 0.23644 | 0.16986 |
| | NeuroSQL | 2.89791 | 0.20532 | 0.20588 |
| 64 | VAE | 10.59103 | 0.25116 | 0.13515 |
| | GAN | 16.09010 | 0.28388 | 0.11962 |
| | NeuroSQL | 2.48879 | 0.19150 | 0.19373 |
| 128 | VAE | 18.35559 | 0.26214 | 0.12317 |
| | GAN | 14.04322 | 0.20137 | 0.17414 |

Table 10: Results on CelebA, a face attributes dataset, using U-Net (unconditional) across latent dimensions. Lower is better for FID (proxy) and LPIPS; higher is better for SSIM.

| Latent dimension | method | FID (proxy) ↓ | LPIPS ↓ | SSIM ↑ |
|---|---|---|---|---|
| 2 | NeuroSQL | 4.96169 | 0.18253 | 0.27537 |
| 2 | VAE | 7.33227 | 0.19269 | 0.25696 |
| 2 | GAN | 14.09393 | 0.22041 | 0.12822 |
| 4 | NeuroSQL | 3.54758 | 0.18932 | 0.27993 |
| 4 | VAE | 4.99991 | 0.19061 | 0.22841 |
| 4 | GAN | 7.66398 | 0.19251 | 0.14899 |
| 8 | NeuroSQL | 3.96161 | 0.19244 | 0.24416 |
| 8 | VAE | 5.48428 | 0.20413 | 0.20573 |
| 8 | GAN | 20.46054 | 0.16577 | 0.18463 |
| 16 | NeuroSQL | 2.72089 | 0.17970 | 0.24505 |
| 16 | VAE | 10.00917 | 0.19001 | 0.18914 |
| 16 | GAN | 14.48838 | 0.18500 | 0.15935 |
| 32 | NeuroSQL | 14.65015 | 0.21850 | 0.27135 |
| 32 | VAE | 31.10957 | 0.24081 | 0.21405 |
| 32 | GAN | 30.37928 | 0.31335 | 0.13607 |

Table 11: Results on AFHQ, an animal faces dataset, using ConvNet across latent dimensions. Lower is better for FID (proxy) and LPIPS; higher is better for SSIM.

| Latent dimension | Method | FID (proxy) ↓ | LPIPS ↓ | SSIM ↑ |
|---|---|---|---|---|
| 2 | NeuroSQL | 22.023664 | 0.563209 | 0.339032 |
| 2 | VAE | 44.276279 | 0.531442 | 0.284219 |
| 2 | GAN | 74.178604 | 0.693773 | 0.080698 |
| 4 | NeuroSQL | 35.913769 | 0.538813 | 0.357857 |
| 4 | VAE | 35.430927 | 0.527985 | 0.285893 |
| 4 | GAN | 48.706165 | 0.669665 | 0.129869 |
| 8 | NeuroSQL | 37.569298 | 0.505324 | 0.353983 |
| 8 | VAE | 83.373650 | 0.516064 | 0.211640 |
| 8 | GAN | 30.621849 | 0.797056 | 0.191322 |
| 16 | NeuroSQL | 52.609642 | 0.564259 | 0.370944 |
| 16 | VAE | 88.436768 | 0.527512 | 0.161437 |
| 16 | GAN | 98.826027 | 0.620977 | 0.066925 |
| 32 | NeuroSQL | 31.695450 | 0.512806 | 0.339188 |
| 32 | VAE | 95.740288 | 0.499635 | 0.172815 |
| 32 | GAN | 46.238804 | 0.751076 | 0.072116 |
| 64 | NeuroSQL | 28.105133 | 0.532103 | 0.347139 |
| 64 | VAE | 127.357986 | 0.525766 | 0.140224 |
| 64 | GAN | 50.671597 | 0.753972 | 0.067132 |
| 128 | NeuroSQL | 17.399908 | 0.591030 | 0.368674 |
| 128 | VAE | 23.283272 | 0.789889 | 0.374535 |
| 128 | GAN | 59.287437 | 0.754374 | 0.042504 |

Table 12: Results on AFHQ, an animal faces dataset, using ConvNet — with results averaged across latent dimensions.

| Method | FID (proxy) ↓ | LPIPS ↓ | SSIM ↑ |
|---|---|---|---|
| NeuroSQL | **32.19** | **0.544** | **0.354** |
| VAE | 71.13 | 0.560 | 0.233 |
| GAN | 58.36 | 0.720 | 0.093 |

Table 13: Results on AFHQ, an animal faces dataset, using ResNet across latent dimensions. Lower is better for FID (proxy) and LPIPS; higher is better for SSIM.

| Latent dimension | Method | FID (proxy) ↓ | LPIPS ↓ | SSIM ↑ |
|---|---|---|---|---|
| 2 | NeuroSQL | 7.267559 | 0.459967 | 0.267403 |
| 2 | VAE | 15.259568 | 0.448991 | 0.235907 |
| 2 | GAN | 39.100639 | 0.550003 | 0.091928 |
| 4 | NeuroSQL | 11.560558 | 0.460007 | 0.246023 |
| 4 | VAE | 13.387914 | 0.432443 | 0.192707 |
| 4 | GAN | 37.873501 | 0.356168 | 0.120893 |
| 8 | NeuroSQL | 13.611592 | 0.471643 | 0.241053 |
| 8 | VAE | 11.736956 | 0.431859 | 0.204782 |
| 8 | GAN | 29.584837 | 0.486513 | 0.081718 |
| 16 | NeuroSQL | 17.987249 | 0.341935 | 0.208179 |
| 16 | VAE | 9.721145 | 0.347468 | 0.178343 |
| 16 | GAN | 17.546219 | 0.483133 | 0.107947 |
| 32 | NeuroSQL | 14.799847 | 0.438891 | 0.255200 |
| 32 | VAE | 8.249439 | 0.341844 | 0.188594 |
| 32 | GAN | 22.425539 | 0.363643 | 0.113281 |
| 64 | NeuroSQL | 10.561233 | 0.558646 | 0.252005 |
| 64 | VAE | 14.458963 | 0.404593 | 0.202796 |
| 64 | GAN | 40.478588 | 0.423956 | 0.107915 |
| 128 | NeuroSQL | 11.019302 | 0.489376 | 0.261104 |
| 128 | VAE | 13.060718 | 0.353215 | 0.208177 |
| 128 | GAN | 18.462259 | 0.381437 | 0.104066 |

Table 14: Results on AFHQ, an animal faces dataset, using ResNet — with results averaged across latent dimensions.

| Method | FID (proxy) ↓ | LPIPS ↓ | SSIM ↑ |
|---|---|---|---|
| NeuroSQL | 12.40 | 0.460 | **0.247** |
| VAE | **12.27** | **0.394** | 0.202 |
| GAN | 29.35 | 0.435 | 0.104 |

Table 15: Results on AFHQ, an animal faces dataset, using U-Net across latent dimensions. Lower is better for FID (proxy) and LPIPS; higher is better for SSIM.

| Latent dimension | Method | FID (proxy) ↓ | LPIPS ↓ | SSIM ↑ |
|---|---|---|---|---|
| 2 | NeuroSQL | 17.626825 | 0.595854 | 0.272737 |
| 2 | VAE | 10.595321 | 0.620105 | 0.279837 |
| 2 | GAN | 23.709433 | 0.405830 | 0.116950 |
| 3 | NeuroSQL | 10.193194 | 0.617973 | 0.278050 |
| 3 | VAE | 29.156761 | 0.578171 | 0.173115 |
| 3 | GAN | 17.098032 | 0.607903 | 0.074634 |
| 4 | NeuroSQL | 10.773172 | 0.561935 | 0.258762 |
| 4 | VAE | 14.463408 | 0.576194 | 0.117610 |
| 4 | GAN | 13.078835 | 0.626831 | 0.057038 |
| 8 | NeuroSQL | 11.994763 | 0.638746 | 0.275091 |
| 8 | VAE | 42.187984 | 0.571152 | 0.062857 |
| 8 | GAN | 75.077110 | 0.618699 | 0.039482 |
| 16 | NeuroSQL | 10.660514 | 0.659894 | 0.266318 |
| 16 | VAE | 51.288338 | 0.550339 | 0.135791 |
| 16 | GAN | 97.593185 | 0.703616 | 0.010010 |
| 32 | NeuroSQL | 16.301731 | 0.595429 | 0.252468 |
| 32 | VAE | 28.342377 | 0.566271 | 0.076232 |
| 32 | GAN | 25.781809 | 0.604119 | 0.025050 |
| 64 | NeuroSQL | 9.855629 | 0.660927 | 0.276445 |
| 64 | VAE | 62.289070 | 0.585000 | 0.093420 |
| 64 | GAN | 99.730621 | 0.702493 | 0.010348 |

Table 16: Results on AFHQ, an animal faces dataset, using U-Net — with results averaged across latent dimensions $\{2, 3, 4, 8, 16, 32, 64\}$.

| Method | FID (proxy) ↓ | LPIPS ↓ | SSIM ↑ |
|---|---|---|---|
| NeuroSQL | **12.486547** | 0.618680 | **0.268553** |
| VAE | 34.046180 | **0.578176** | 0.134123 |
| GAN | 50.295575 | 0.609927 | 0.047645 |

On average, NeuroSQL achieves the best FID (proxy) and SSIM; VAE attains the lowest LPIPS.

Table 17: Ablation results on MNIST, a database of handwritten digits (all runs). Lower is better for FID (proxy) and LPIPS; higher is better for SSIM.

| Latent dimension | Seed | Method | FID (proxy) ↓ | LPIPS ↓ | SSIM ↑ |
|---|---|---|---|---|---|
| 2 | 11 | NeuroSQL | 0.696835 | 0.039503 | 0.564344 |
| 2 | 11 | VAE | 1.070473 | 0.058065 | 0.200167 |
| 2 | 11 | GAN | 2.157639 | 0.054278 | 0.245780 |
| 3 | 11 | NeuroSQL | 0.527451 | 0.030257 | 0.668574 |
| 3 | 11 | VAE | 1.443453 | 0.060009 | 0.157831 |
| 3 | 11 | GAN | 1.849820 | 0.057282 | 0.220785 |

Table 18: Results on MNIST, a database of handwritten digits — averaged over latent dimensions $\{2, 3\}$ (seed = 11).

| Method | FID (proxy) $\downarrow$ | LPIPS $\downarrow$ | SSIM $\uparrow$ |
|--------|--------------------------|--------------------|------------------|
| NeuroSQL | **0.612143** | **0.034880** | **0.616459** |
| VAE | 1.256963 | 0.059037 | 0.178999 |
| GAN | 2.003730 | 0.055780 | 0.233283 |

NeuroSQL is best on all three metrics (FID (proxy), LPIPS, and SSIM).

