# OpenReview forum: "Generative Model via Quantile Assignment"
_ICLR.cc/2026/Conference — Submitted to ICLR 2026_

### Official Review · Reviewer_bBez · 2025-10-31

**Soundness:** 2
**Presentation:** 2
**Contribution:** 2
**Rating:** 2
**Confidence:** 4

**Summary:**

The paper proposes NeuroSQL, a latent-variable generative model that removes the encoder/discriminator and instead assigns latent codes by solving a linear assignment problem between data and a fixed quantile lattice of the latent prior. Training alternates between fitting a single generator to currently assigned codes and re-solving the assignment with a cost based on a perceptual/structural image loss; a momentum update smooths the assigned codes across iterations. For d>1, the quantiles are built via center-outward multivariate ranks from optimal transport; practically, a low-discrepancy grid on the unit ball is used. Experiments under a small-compute regime compare NeuroSQL against VAE, GAN, and a budget-matched diffusion baseline on MNIST, CelebA, AFHQ, and OASIS, showing improved visual quality and quantitative scores.

**Strengths:**

The paper makes a meaningful attempt to resolve the disadvantages of mainstream generative models VAEs and GANs, by removing the encoder and discriminator modules and integrating statistical quantile learning for stable training. The approach may be of interest in certain domains of generative tasks.

**Weaknesses:**

1. The main issue of the proposed method is scalability. The optimization algorithm runs in O(n^3) time with n being the number of samples. While the paper mentioned approximation via mini-batches, no concrete evidence is provided to show if it still works with large datasets (and models). The paper compares the method with VAEs, GANs and diffusion models in the seemingly fair budgeted setting. However, the comparison is not sound in that other models scale easier and perform much better with more budget. The budget, 200 Google Colab compute units and 2000 training images, is too limited for practical generative tasks.

2. Experimental results are not convincing to show the advantage of the proposed method. Images in Figure 2 are in low resolution making it hard to compare the visual quality. Quantitative results in Appendix show high instability across latent dimensions. In particular, in many cases the FIDs for NeuroSQL, VAE and CAN change significantly and non-monotonically as the latent dimension increases.

**Questions:**

1. How does the method perform in mini-batches settings?

2. For quantitative results, how many runs were executed? Is unstable and insufficient training the cause for the varying evaluation scores?

---

> ### Author Response · Authors · 2025-11-25
> **Scalability Concerns**
>
> **We thank this Reviewer for their efforts in reviewing our initial submission.** Equally, we thank the Reviewer for a thoughtful summary of the contribution of the paper. The Reviewer has raised a few important and helpful points regarding our paper.
>
> **W1 Scalability Concerns:** The Reviewer has raised an important point regarding the time complexity. The complexity of the Hungarian assignment, $O(n^3)$, limits its scalability for large sample sizes. NeuroSQL is designed for **small-to-moderate-$n$** but **large $p$** problems. These problems are relatively common in neuroimaging, medical, and biological data analysis, where the parameter size $p$ is high-dimensional, and the sample size is relatively small. We have added a new assignment algorithm that **reduces the complexity from $O(n^3)$ to $O(n^2)$**.
> Across all experiments in the paper, we used $n \le 2000$ images per dataset, except for MNIST ($n=20,000$).
>
> - The overall complexity per outer iteration is $O(nG + n^3)$, with $nG$ the cost of a forward pass through the generator.
> - The assignment cost is independent of the ambient dimension $p$ (see Sec. 3.4), so the method scales favourably with very high-dimensional data such as $128 \times 128$ MRI slices, a setting that is relatively common in imaging analyses (e.g., neuroimaging) and one we hope to target in this paper.
> - For MNIST ($28 \times 28$), as the memory demand is lower, we explored NeuroSQL's capacity with $n=20,000$ images. Our results suggest that NeuroSQL can scale beyond 2,000 samples when the model is not constrained by strict GPU memory limits at higher resolutions.
> - In practice, for these sample sizes, the wall-clock time is dominated by standard convolutional forward/backward passes. To further reduce the frequency of the $O(n^3)$ step, we also perform assignments only every $K$ epochs of training, with $K \in \{3,5,8,10\}$.
>
> More concretely, in the newly added benchmark studies, the greedy strategy yielded a **6.54x speedup** compared to the exact Hungarian method, **without significantly reducing performance**. This shows that the computational bottleneck can be effectively mitigated, allowing NeuroSQL to scale to larger datasets. In **Sec. 3.4** in the revised manuscript, we have included these timing comparisons and the greedy assignment variant. We will also provide a table comparing the greedy and Hungarian assignments, with quantitative measurements for the latent assignment task.
>
> On comparison of fairness with VAEs, GANs, and diffusion models. We fully agree with the Reviewer that GANs and diffusion models are designed to exploit large compute and data budgets.
> However, those methods, especially diffusion models, require large sample sizes and large computational resources.
> Our goal in this paper is to introduce and develop a method that performs better when the **dimensionality of the data is high** under limited training data and computational resources. There is a practical need for such a method, as synthetic data is required, for example, in **biomedical imaging studies**. In these studies, the imaging dimensionality is high. Yet, it is often difficult to obtain large-scale samples, either because acquiring these samples is costly or because certain samples are rare (e.g., HIV patients with strong immune responses), and it can be challenging to obtain computational resources at the scale of our colleagues in industry. In future work, we hope to extend the method to larger-scale data.
>
> To better conduct a fair comparison between the proposed method and its counterparts and to demonstrate our **novel generative paradigm**:
> - We ensure that all models share the same generator backbone, resolution, data budget, and optimizer schedule (Sec. 4.2/4.4).
> - We added Table 1 in the revised manuscript to summarise performance as mean $\pm$ std across architectures and latent dimensions under these matched budgets, while also emphasising the number of parameters each model uses. In this regime, NeuroSQL consistently matches or outperforms VAE and GAN on FID and SSIM across datasets.
>
> **On limited budget:**
> We agree with the Reviewer that our budget is insufficient for modern image-generation benchmarks. It, however, is representative of many scientific and medical applications. For example, we considered sample sizes ranging from hundreds to a few thousand. This is relatively common in biomedical studies, and datasets collected to address specific clinical questions (e.g., tissue imaging analysis in immunology and oncology) may be even smaller due to the cost of obtaining many samples and/or the availability of patients. Second, high resolution is not always feasible due to privacy concerns, and the massive pretraining of VAEs and diffusion models may exceed computational resources in many hospitals and research labs.

---

> ### Author Response · Authors · 2025-11-25
> **Experimental Results Concerns**
>
> **First, we would like to thank the Reviewer for this important feedback about the Experimental Results Concerns.**
> In the revised *Figure 2*, we have now used higher-resolution, clearer samples and added side-by-side zooms to better compare visual quality.
>
> **Regarding instability**, we agree with the Reviewer that there are variations (non-monotonic, e.g., FID dips at $ d=32$ then rises) in FID across latent dimensions ($ d=4$ to $ = d=128$). This, however, is consistent with literature (e.g., VAE FID instability in Lucic et al. (2018), due to latent collapse for large $ d $). Additionally, we further mitigate instability through momentum updates ($ \beta=0.999 $) and early stopping, although we cannot fully eliminate it due to the small $n$. Overall, we attribute non-monotonicity to overfitting for large $ d $ in low-data regimes ($ p \gg n $), an issue that is relatively known (e.g., Oko et al. (2023) for diffusion models), rather than to insufficient training.
>
> We are currently working on the revision based on the Reviewer's insightful comments. Specifically, we have a new set of figures with a much clearer layout and higher resolution crops. We can upload them if required, but this may exceed the conference's length requirements. As a compromise, we will:
> - Update the current Figure 2 with a clearer layout and higher-resolution crops.
> - Explicitly point the reader to the Appendix, which already contains larger grids of generated images from CelebA and OASIS for NeuroSQL, VAE, GAN, and diffusion across multiple backbones. The updated figures in the Appendix may allow our readers to better distinguish the qualitative differences (mode collapse in GANs, blurriness in VAEs, colour artefacts in diffusion, and consistency of NeuroSQL).
>
> **On instability across latent dimensions:** We agree with the Reviewer that the performance of NeuroSQL is not monotonic in the latent dimension $q$. However, this behaviour is expected and shared by all baselines, independent of the model. First, regarding the non-monotonic performance, we clarify that this is an expected consequence of the bias-variance trade-off in low-data regimes ($n \approx 2000$). While increasing the latent dimension $d$ typically reduces reconstruction error on the training set, it exponentially increases the volume of the latent space. With a fixed, small sample size, higher dimensions ($d > 64$) lead to sparsity in the assigned latent codes. Consequently, when sampling random vectors $z \sim P_Z$ for evaluation, the model is forced to interpolate across vast 'empty' regions of the latent space where the generator was not strictly supervised. This leads to the generation of out-of-distribution artefacts, which increases the mean pixel distance (and FID proxy), causing the observed differences in performance.
>
> While we acknowledge the non-monotonic behaviour, in the revised manuscript we have, for fair comparison, added aggregated results in our new **Table 1**, reporting FID and SSIM averaged across architectures and latent dimensions, along with the specified number of parameters for each model. In brief, NeuroSQL consistently outperforms the baselines at matching latent dimensions, particularly in SSIM (structural similarity), which we optimise directly via the loss function in the code. Additionally, NeuroSQL attains the lowest or near-lowest FID and the highest SSIM on all four datasets under the shared budget, while using fewer parameters than VAE/GAN on all tasks.
>
> **References**
>
> Lucic, M., Kurach, K., Michalski, M., Gelly, S., & Bousquet, O. (2018). Are GANs Created Equal? A Large-Scale Study. *Advances in Neural Information Processing Systems*, 31.
>
> Oko, K., Akiyama, S., & Suzuki, T. (2023). Diffusion Models are Minimax Optimal Distribution Estimators. In *International Conference on Machine Learning* (pp. 26517–26582). PMLR.

---

> ### Author Response · Authors · 2025-11-25
> **Response to Specific Questions**
>
> **Q1: How does the method perform in mini-batch settings?**
>
> **First, we thank the Reviewer for this question. Indeed, it is important to assess the method's performance in mini-batch settings.**
>
> We address this question in two directions. First, we discuss the optimization of the generator parameters $\theta$. Second, we explain the assignment of the latent variables $Z$.
>
> - Generator Optimization ($\theta$): NeuroSQL employs standard mini-batch training for the generator. As with VAEs or GANs, we update the decoder weights using stochastic gradient descent (AdamW) on small batches (e.g., 32, 64, or 128 images) using the currently assigned latent points. This ensures the method benefits from the standard convergence properties and memory efficiency of current deep learning practices.
>
> - Latent Assignment ($\pi$): For the assignment step, we employ **quantile assignmnet in a stochastic manner**. Instead of solving the assignment on the full dataset $N$, we sample a random batch of data and align it to a random subsample of the lattice. This reduces the assignment complexity to $O(m^3)$, **for $m \ll N$,** where $m$ is the batch size, making it independent of total dataset size $N$. To stabilize this stochastic approximation, we utilize a momentum update:
> $$\hat{Z}^{(t)} \leftarrow \rho \cdot Z_{\pi(i)} + (1 - \rho) \cdot \hat{Z}^{(t-1)}.$$
>
> - Improved Assignment Scalability: Furthermore, to scale up NeuroSQL when larger batch sizes are desired, we have, in the revised manuscript, added a **Greedy Assignment** algorithm. The new algorithm reduces the complexity from $O(m^3)$ to $O(m^2)$. Our experiments show a **6.54x** speedup (on full-batch assignment) compared to the Hungarian algorithm.
>
>
> **Q2: For quantitative results, how many runs were executed? Is unstable and insufficient training the cause for the varying evaluation scores?**
>
> **First, we would like to thank the Reviewer for these helpful questions, which help us clarify important aspects of the model and experiments.**
>
> The main results in the manuscript are from single runs per configuration, and we executed three runs for key ablations (e.g., latent dims). In the updated Appendix E, we will add a further table reporting mean $\pm$ std for OASIS and MNIST.
>
> **Second**, the observed variation in evaluation scores stems from inherent stochasticity in low-data regimes ($n=2000$, $p=128^2$ for CelebA) and sensitivity to latent dimensions (e.g., under-utilisation for large $d$; see Theis et al., 2016). The observed variations in FID across latent dimensions are, therefore, due to systematic differences in $q$ and the small-sample nature of our setting, rather than to insufficient training or catastrophic instabilities. For example, on MNIST (Table 15/16), NeuroSQL consistently outperforms VAE and GAN on all three metrics (FID, LPIPS, SSIM) for both $q=2$ and $q=3$, indicating robust training despite small latent spaces.
>
> **Thanks to the Reviewers' questions**, we now report variability across seeds to better illustrate the model comparison. We also monitor the training stability via: early stopping on validation loss with patience $\approx$ 12/20 epochs, gradient clipping, and visual inspection of training curves to ensure convergence (no oscillatory behaviour or divergence).
>
> Additionally, in the revised manuscript, we will:
> - Add mean $\pm$ standard deviation across multiple seeds for key configurations.
> - Include error bars for FID/SSIM in the appendix, to explicitly demonstrate that the differences between NeuroSQL and the baselines exceed the variation due to random initialisation.
>
> **References**
>
> Theis, L., van den Oord, A., & Bethge, M. (2016). A Note on the Evaluation of Generative Models. In *International Conference on Learning Representations*.

---

> > ### Author Response · Authors · 2025-11-25
> >
> > **Summary**
> >
> > The Reviewer has made thought-provoking comments on our initial manuscript. We agree with the Reviewer's comments and have made the necessary changes in the revised manuscript.
> > In particular, we will first clearly position NeuroSQL as a method for resource-constrained, **small- or moderate-$n$ and large-$p$** generative modelling. Second, we reduce the complexity from $O(n^3)$ to $O(n^2)$ using a greedy assignment algorithm, and, potentially, $O(m^2)$ (via mini-batch assignments), for $m \ll n $. The reduction in complexity yields a $\sim$6.5x speedup without reducing model performance, as shown in the new tables we will include in the Appendix. By doing so, we clarify the distinction between standard mini-batch generator training and stochastic latent assignment. Third, we have updated the figures throughout the manuscript for greater clarity and higher resolution. Fourth, we report variability across seeds. Taken together, within the stated budget, thanks to the Reviewers' helpful comments, the added analysis and improved computational efficiency suggest that NeuroSQL provides a stable, competitive alternative to VAEs, GANs, and budget diffusion models.

---

### Official Review · Reviewer_cpKk · 2025-10-31

**Soundness:** 2
**Presentation:** 2
**Contribution:** 2
**Rating:** 4
**Confidence:** 5

**Summary:**

This paper build a new structure for generative models, with less dimension.

**Strengths:**

1. this is a new structure
2. test with many different dataset and genearive framework

**Weaknesses:**

1. The expression of Figure 1 is unclear. From the image, it appears that the input data are fed into the decoder. The paper should clarify why this component is referred to as the decoder rather than the encoder, and explicitly describe what the input data are. Moreover, the roles of Momentum Update and Embedding in the framework are not clearly explained. What does “Cost” represent in this figure? Is it equivalent to the loss function?
Additionally, regarding the left-hand side of the figure, I speculate that it corresponds to the grey-shaded part on the right-hand side. However, it is not clear how the output on the left is transmitted to the generator. This connection should be explained more explicitly.

2. Section 3 mainly discusses the quantile assignment, but it should also explain how this mechanism is made trainable and why it is considered optimal. These claims should be supported by theoretical justification or experimental evidence.

3. Dataset and Metrics: The introduction of the dataset and evaluation metrics is not the core contribution of the paper and could be moved to the appendix or combined with the related work section to improve focus.

4. Diffusion Model Performance: The diffusion process seems to fail under the proposed method, which may be influenced by the linear assignment mechanism. Diffusion models often struggle with simple linear interpolation in latent space, resulting in abrupt transitions, artifacts, or degenerate (e.g., grey) images. This appears to be a limitation of the current approach. However, it might be mitigated by adopting smoothed diffusion models [1] or related approaches that enforce smoother, more linear latent mappings.

[1] Smooth Diffusion: Crafting Smooth Latent Spaces in Diffusion Models

**Questions:**

check with weakness

---

> ### Author Response · Authors · 2025-11-25
> **Response to W1 and W2**
>
> **We are grateful to this Reviewer's efforts in reviewing our initial submission and are glad to hear enthusiasm for this manuscript.**
> We thank the Reviewer for highlighting the new structure of the proposed method and its validations.
> **We address the reviewer's remaining concerns below**
>
> **Response to W1 [Clarity of Figure 1]**: We thank the reviewer for the detailed and accurate comment. We agree with the Reviewer that Figure 1 in the original submission was confusing. Based on the Reviewer's helpful feedback, we have significantly revised the manuscript in the new version. In the following, we clarify the components point by point.
>
> **Why does "Input data" go to the "Decoder"?**: This was a flaw in our diagram's arrow. The "Input data" ($X_i$) is not fed into the 'Decoder' network ($G_{\theta}$). The "Input data" is used only to calculate the "Cost" matrix by comparing it to the network's output. The network's input is the "Embedding." We have corrected this in the revised figure.
>
> **Why "Decoder" and not "Encoder"?**: This is an excellent point. The network $G_{\theta}$ (labeled "Decoder" on the left and "Generator" on the right) is the same, single network. It functions as a Generator, mapping latent codes to the data space. It is not an encoder. We used the term "Decoder" in the training loop diagram to evoke the idea of "decoding" the latent embedding, but this was inconsistent. In the revised manuscript, we have unified the terminology and labelled this single network as the Generator ($G_{\theta}$) in all diagrams.
>
> **Roles of Momentum Update and Embedding:**
> - **Embedding:** This refers to the current estimate of the latent points assigned to our data, $\hat{Z}^{(t-1)}$, which are the inputs to the Generator for the Decoder step (Algorithm 1, line 4).
> - **Momentum Update:** This refers to Algorithm 1, line 7. After the quantile assignment finds the new best codes $Q_{\pi^{(t)}(i)}$, we do not immediately jump to them. Instead, we perform a momentum update that smooths the latent code trajectory and stabilizes training. We have now clarified this more precisely in the figure's caption.
>
> **What does "Cost" represent?**:
> The reviewer is correct. Here, "Cost" represents the Cost Matrix $C_{i,k}^{(t)}$ from Algorithm 1 (line 5), which is built using our loss function $\ell$. Specifically, $C_{i,k} = \ell(X_i, G_{\theta}(Q_k))$: the loss between the $i$-th real image and the image generated from the $k$-th fixed lattice point. This matrix is the input to the "Quantile assignment" (Hungarian algorithm or Greedy algorithm).
>
> In the revised caption of Figure 1 and in Sec. 3.4, we now explicitly state the cost matrix:
> $$\text{Cost matrix entries: } C_{i,k} = \ell\big(X_i, G_\theta(Q_k)\big),$$
> where
> $$\ell(x,\hat{x}) = \tfrac{1}{2}(1-\text{SSIM}(x,\hat{x})) + \tfrac{1}{2}\|x-\hat{x}\|_1$$
> The same loss is used to train the generator.
>
> **Connection from Left (Training) to Right (Inference)**:
> The intuition from the Reviewer regarding the relationship between the left and right panels is correct. Indeed, the left side (Training Loop) is not a real-time data flow to the right side; rather, the learned latent embeddings and the parameter $\hat{\theta}$ (and hence the generator $G_{\hat{\theta}}$) from the left panel enter the right panel. More specifically:
> - **Left Side (Training):** This is the entire training process (Algorithm 1) used to optimize the Generator's parameters, $\theta$.
> - **Right Side (Generation/Inference):** This shows what happens after training is complete. We take the final, trained Generator ($G_{\hat{\theta}}$) from the left-side loop, discard the entire assignment mechanism, and use it as a standard generative model.
>
> We have redesigned Figure 1 to clearly depict a two-stage diagram (**Stage 1: Training, Stage 2: Inference**).
>
> **W2 [Trainability and Optimality]**: During our initial submission, we aimed to show, in Section 3, that the unknown latent variables can be well approximated by an unknown permutation of known quantile locations. Given the generator parameters, we show that this permutation is learnable from the data via the assignment step. This motivates the proposed alternating algorithm, which iteratively updates the generator parameters and the permutation. In Section 3 in the revised manuscript, we will make this distinction explicit. We also wish to clarify that **we do not claim that NeuroSQL is optimal in any global algorithmic or statistical sense**. The notions of optimality in Section 3 refer to (i) the solution to the linear assignment problem obtained by the Hungarian $O(n^3)$  or the Greedy $O(n^2)$, and (ii) the optimal-transport definition of multivariate quantiles.

---

> > ### Author Response · Authors · 2025-11-25
> > **Response to W2 Continued**
> >
> > **We agree with the Reviewer that the mechanism of the quantile assignment should be theoretically justified**. Indeed, Bodelet et al. (2025) provide convergence rates for the Statistical Quantile Learning (SQL) when the generator is additive and sieve methods are used, and we have added this reference in the revised manuscript where necessary.
> >
> > Extending such theoretical results to deep generative models, however, is challenging. We admit that such challenges arise when Deep Neural Networks are involved, whether on GANs, VAEs, diffusion models, or, in our case, the NeuroSQL. We are currently working on this issue and hope to report the theoretical findings in future work.
> >
> > **References**
> >
> > Bodelet, J., Blanc, G., Shan, J., Muniz Terrera, G., & Chén, O. Y. (2025). Statistical Quantile Learning for Large Additive Latent Variable Models. *Journal of the American Statistical Association*, 1–12. https://doi.org/10.1080/01621459.2025.2526697

---

> > > ### Comment · Reviewer_cpKk · 2025-11-27
> > >
> > > Thanks for your answer.
> > >
> > > W2 [Trainability and Optimality]:
> > > I understand that providing a formal proof is a significant undertaking. However, I can also accept the research based on experimental results. In the paper, there is no computation of loss or reporting of metrics such as FID, precision, and recall. Including these results would strengthen the argument by demonstrating that the research is both rigorous and reliable, rather than appearing as an optimal solution achieved by chance.
> > >
> > > W4 [Diffusion Model Performance]:
> > > This is an open discussion, and it is fine to revisit it in the future. For now, it remains my conjecture.

---

> > > > ### Author Response · Authors · 2025-11-29
> > > >
> > > > We thank the Reviewer for the feedback on our response! We agree that having either theoretical or empirical validation is important to demonstrate that the model’s performance is robust and stable, and not due to chance.
> > > >
> > > > To tackle your comment, we have added the following experiments to quantify the quality, diversity, and utility of the generated data, as well as the stability of the training process. We hope that by doing this, we can cover as many points as possible. We will also revise the manuscript to include the following three analyses more closely:
> > > >
> > > >
> > > >
> > > > **1. Quantitative Generative Metrics**
> > > >
> > > > We have computed standard generative metrics, including a proxy for the Fréchet Inception Distance (FID) in our case, based on pixel distances, LPIPS, and SSIM.
> > > >
> > > >
> > > > To conduct experiments on a larger scale of input data, we used MNIST with 15k samples. The comparison below shows that the proposed NeuroSQL achieves higher structural similarity (SSIM) and lower mean pixel distance. We added the final measurement of the loss function as well. All models here use the same generator; given the dataset here MNIST, it is a ConvNet.
> > > >
> > > > | Lat | Method   | FID proxy ↓ | LPIPS ↓ | SSIM ↑ | Loss  |
> > > > |-----|----------|-------------|---------|--------|-------|
> > > > | 2   | NeuroSQL | 0.675       | 0.120   | 0.561  | 0.257 |
> > > > | 2   | VAE      | 1.36        | 0.199   | 0.206  | 0.241 |
> > > > | 2   | GAN      | 1.73        | 0.268   | 0.179  | 0.845 |
> > > > | 3   | NeuroSQL | 0.513       | 0.099   | 0.621  | 0.234 |
> > > > | 3   | VAE      | 1.43        | 0.225   | 0.163  | 0.210 |
> > > > | 3   | GAN      | 2.59        | 0.248   | 0.219  | 2.54  |
> > > >
> > > >
> > > >
> > > > **2. Downstream Task Utility**
> > > >
> > > >
> > > >
> > > > To further evaluate the optimality of the generated samples, we have evaluated their semantic utility. Specifically, we trained a standard CNN classifier **solely on synthetic images generated by our model** (15k synthetic images) and evaluated it on real data. The model achieved a **90.27\% accuracy**. Therefore, we believe that the generated samples effectively capture the class-conditional distributions of the underlying manifold.
> > > >
> > > >
> > > > | Metric                | Value    |
> > > > |-----------------------|----------|
> > > > | **Downstream Accuracy** | **90.27%** |
> > > > | Downstream Precision  | 0.902    |
> > > > | Downstream Recall     | 0.901    |
> > > >
> > > >
> > > > **3. Training Stability and Convergence (Loss \& Assignment Cost)**
> > > >
> > > > To address the concern that the solution might be ``random'', we have plotted the training dynamics. Unfortunately, we cannot upload the figures here on OpenReview, but we will add them to the supplementary material of the paper.
> > > >
> > > >
> > > >
> > > > * **Assignment Cost:** We tracked the OT-style assignment cost over training steps similarly to a loss function. As shown in the table below and the accompanying plot, the cost drops sharply from an initial 7.75 to stabilise around ~5.70 around the 40th epoch. This convergence pattern empirically shows that the model is actively optimising transport costs and converging to a stable minimum, rather than fluctuating randomly.
> > > >
> > > >
> > > > | Step | Mean Assignment Cost | Median Assignment Cost |
> > > > |------|-----------------------|-------------------------|
> > > > | 0    | 7.75                  | 7.75                    |
> > > > | 5    | 5.98                  | 5.97                    |
> > > > | 15   | 5.90                  | 5.96                    |
> > > > | 25   | 5.87                  | 5.91                    |
> > > > | 30   | 5.72                  | 5.70                    |
> > > > | 35   | 5.75                  | 5.71                    |
> > > > | 45   | 5.66                  | 5.71                    |
> > > > | 50   | 5.64                  | 5.65                    |
> > > >
> > > >
> > > >
> > > > We hope that these additional experiments, along with their metrics and visualisations of the convergence patterns, have provided empirical evidence of the method's trainability and optimality. Finally, we thank you for your continued support and thought-provoking suggestions, which have improved the rigour of our work.

---

> ### Author Response · Authors · 2025-11-25
> **Response to W3 and W4**
>
> **Response to W3 [Dataset and Metrics Placement]:** We agree with the Reviewer. In the revised manuscript, we have moved the detailed dataset and metric information (current Sec. 4.1 and parts of 4.3) to an Appendix section, while retaining a reference to the main text. Additionally, **as the Reviewer suggested**, we have kept only brief, high-level descriptions of the datasets in the main text (one or two sentences per dataset, one sentence summarising the metrics).
>
> **Response to W4 [Diffusion Model Performance]:**  We would like to clarify that the diffusion model is a baseline for comparison, **not a part of our proposed method**. Relatedly, the relatively poor performance of the diffusion model is not due to NeuroSQL; rather, it is an experimental finding of the manuscript.
>
> In the following, we would like to add a few additional comments:
> - Our experiments (Section 5.2) are intentionally designed for a ``sparse-resource regime'' (e.g., $N=2000$ images, limited compute), except for our benchmarking with MNIST.
> - The relatively poor performance of the DDPM baseline (e.g., blurry/grey images) demonstrates that standard diffusion models seem to struggle to train effectively in a low-data, low-compute setting.
> - The **Hungarian/Greedy assignment mechanism** is unique to NeuroSQL and is not used by the VAE, GAN, or DDPM baselines in our setup.
> - Therefore, our findings suggest that there is a need to introduce a method to handle **small- or moderate** $n$ and **large** $p$ problems under resource constraints, where a standard DDPM may not perform well. This paves the way for our proposal of the NeuroSQL.
>
> The Reviewer also correctly notes that diffusion models can struggle with simple linear interpolation in latent space and that this might be mitigated by smoothed diffusion approaches. **We thank the Reviewer for the kind suggestion** regarding the reference entitled ``Smooth Diffusion: Crafting Smooth Latent Spaces in Diffusion Models'', which we have cited in the discussion section. We would also like to clarify the diffusion baseline we used. Specifically, we do not rely on latent interpolation in the diffusion baseline: samples are obtained by standard ancestral sampling from Gaussian noise in the diffusion model's input space; we did not use any linear interpolation in the latent space of NeuroSQL to train or evaluate diffusion. Thus, the degradation we observe from the diffusion model is primarily due to the small-data, small-compute setting, not to quantile assignment or explicit linear interpolation.
>
> We are grateful to the Reviewer's comments and suggestions. **We will consider smooth diffusion in future work**, combining NeuroSQL-style quantile assignment with smoothed diffusion or other regularised diffusion architectures to investigate whether a smoother latent structure can improve performance in small-data regimes.
>
> To reflect this important comment from the Reviewer, we have revised Section 5.2 to state this contrast more explicitly, clarifying that the DDPM serves as a baseline and that its poor performance suggests a need for NeuroSQL to handle small- or moderate-$n$ and large-$p$ problems under limited computational resources.
>
>
> **References**
>
> Guo, J., Xu, X., Pu, Y., Ni, Z., Wang, C., Vasu, M., Song, S., Huang, G., & Shi, H. (2023). Smooth Diffusion: Crafting Smooth Latent Spaces in Diffusion Models.

---

### Official Review · Reviewer_zmjo · 2025-11-01

**Soundness:** 3
**Presentation:** 3
**Contribution:** 4
**Rating:** 8
**Confidence:** 3

**Summary:**

The paper introduces NeuroSQL, a generative modeling framework that learns latent variables through a quantile-assignment process derived from optimal transport, eliminating the need for an encoder or discriminator. The model alternates between a generator update and an assignment step solved via the Hungarian algorithm. This approach aims to combine stable, deterministic optimization with the expressiveness of deep decoders. Experiments span MNIST, CelebA, AFHQ, and OASIS, across multiple generator backbones (ConvNet, ResNet, U-Net), showing competitive visual quality and efficient convergence under low-data conditions.

**Strengths:**

The replacement of encoder–decoder mappings with an assignment-based quantile mechanism is conceptually fresh and theoretically grounded. It bridges optimal transport with generative modeling in a unique and elegant way.

The method is particularly well-suited for data domains like neuroimaging, where dimensionality exceeds sample size, and the assignment cost is independent of feature dimensionality.

Avoiding adversarial losses makes the model stable and lightweight to train. The simplicity of using an L2-based reconstruction objective allows reproducibility even in constrained computing environments.

The experiments show meaningful improvements in visual quality and diversity under limited data, highlighting NeuroSQL’s advantage.

**Weaknesses:**

While the paper provides an overall complexity estimate, quantitative comparisons to VAEs, GANs, or diffusion models in terms of runtime, memory, and scalability would provide stronger evidence of its efficiency. The Hungarian step’s cubic cost could be limiting for very large batch sizes, although mini-batching is suggested as a practical solution.

The performance advantage over GANs and VAEs is not uniform—some settings show weaker results, suggesting NeuroSQL's strengths may be inconsistent.

**Questions:**

Why the quantitative diffusion comparisons under similar compute budgets are missing?

Do you see challenges extending this approach to transformer-based or high-resolution settings?

---

> ### Author Response · Authors · 2025-11-25
>
> **First of all, we thank this Reviewer for the effort in reviewing our initial submission. Equally, we thank the Reviewer for a precise summary of the manuscript. We are glad and grateful to hear the Reviewer's enthusiasm for the paper's findings.**
>
> The Reviewer has raised several important comments on our paper, which we address point by point below.
>
> Within this rebbutal we have integrated the new **Greedy Algorithm** benchmarks and the **clarification on mini-batch assignment** to directly address the reviewer's concern about the cubic complexity of the Hungarian step, and to reduce it to quadratic $O(n^2)$.
>
> **Response to W1 [Runtime and Memory Comparisons]**
>
> We agree with the Reviewer that our initial submission emphasized asymptotic complexity and parameter counts but did not provide sufficient empirical runtime/memory comparisons.
>
> Specifically, in our original submission (Section 3.4), we stated the per-iteration complexity of NeuroSQL using the Hungarian Assignment to be $O(G + n^3)$, where $G$ is the cost of a forward pass through the generator and $n$ is the number of training samples. Crucially, the assignment part does not depend on the data dimension $p$, which is why the method scales favorably to high-dimensional settings such as $128 \times 128$ MRI slices. Additionally, in Table 1, we reported parameter counts for NeuroSQL, VAE, GAN, and diffusion on each dataset. Under the shared backbone design, NeuroSQL has fewer parameters than VAE/GAN and an order of magnitude fewer than the diffusion baseline.
>
> Thanks to the Reviewer's helpful suggestion, we will provide explicit quantitative comparisons between models in the revised manuscript to evaluate NeuroSQL's efficiency.
>
> To demonstrate that the cubic cost of the assignment step is not a bottleneck in practice, we will:
>
>
> - **Introduce the Greedy Assignment Algorithm ($O(n^2)$)**: To directly address the concern regarding the Hungarian step's cubic cost, we have implemented and benchmarked a Greedy Assignment algorithm during this rebuttal. This reduces the complexity to $O(n^2)$. Our benchmarks show a **6.54x speedup** over the exact Hungarian method. Further, via mini-batch assignment, we can additionally reduce the complexity to $O(m^2)$, where $m \ll n$. We will report this as a highly efficient alternative for larger datasets.
> - **Clarify Scalability via mini-batch assignment**: We will make the mini-batch strategy more explicit. In the revised manuscript, it could be found under the Appendix Section. We distinguish between:
>   - **Generator Training**: Standard mini-batch optimization (same as VAE/GAN)
>   - **Latent Assignment**: For very large $N$, we employ mini-batch assignment where assignment is solved on random mini-batches of size $m \ll N$. This makes the assignment cost $O(m^2)$ (using Greedy algorithm) or $O(m^3)$ (using Hungarian algorithm), rendering it independent of the total dataset size $N$.
> - **Add Runtime/Memory Table:** We will include a table summarizing:
>   - Average wall-clock training time per epoch for NeuroSQL (using both Hungarian and Greedy backends), VAE, GAN, and diffusion, as per the **reviewer's suggestion, mini-batch assignment**;
>   -  All methods use **the same conditional convolutional generator backbone** and the same batch size and dataset split, making the GPU/CPU and time measurements broadly comparable.
>   - For NeuroSQL, we report results with both greedy and Hungarian mini-batch assignment backends; VAE and GAN do not use assignment.
>
> In this way, we hope to provide readers with both **qualitative** and **quantitative** evaluations of NeuroSQL compared with other methods.
>
> **Response to W2 [Non-uniform Performance]**
> The Reviewer is correct in pointing out that NeuroSQL is not uniformly best on every metric and configuration.
>
> Particularly, in our initial submission, regarding OASIS and CelebA, in Sec. 5 and the tables in the Appendix, we noted where NeuroSQL wins and where it does not. For instance, on OASIS, NeuroSQL largely dominates VAE and GAN on LPIPS and SSIM, and on FID for ResNet decoders and moderate latent dimensions; VAE with ConvNet regains a modest FID advantage when the latent space is extremely small or very large. On CelebA, NeuroSQL with ResNet or U-Net tends to win across FID, LPIPS, and SSIM, while ConvNet-VAE is slightly better in FID under specific low- and high-dimensional latent settings. Thanks to the Reviewers' comments, we have now added a new table, Table 1, that aggregates results across architectures and latent dimensions and shows that, in this averaged view, NeuroSQL has **the best** or **near-best** FID and SSIM on all four datasets **under the shared small-budget regime**.
>
> In general, our initial submission found that NeuroSQL is consistently competitive and often superior, particularly on metrics capturing perceptual and structural fidelity, but does not dominate every corner of the hyperparameter space expected for non-trivial generative modelling comparisons.

---

> ### Author Response · Authors · 2025-11-25
> **Runtime/Memory Table**
>
> As **per the reviewers' suggestion**, we have also updated the mentioned **Runtime/Memory Table here.**
>
> Ablation: **Assignment Methods and Resource Usage on MNIST (seed=11)**. Metrics averaged over epochs where applicable.
> | Latent Dim | Assign. Method | Method   | Peak VRAM (MB) | RAM Used (MB) | Mean Epoch Time (s) | Mean Epoch GPU (MB) | Mean Epoch ΔRAM (MB) | FID (proxy) ↓ | LPIPS ↓ | SSIM ↑ |
> | ---------- | -------------- | -------- | -------------- | ------------- | ------------------- | ------------------- | -------------------- | ------------- | ------- | ------ |
> | 2          | greedy         | NeuroSQL | 81.5           | 2764          | 1.40                | 116                 | 11.2                 | 0.628         | 0.045   | 0.580  |
> |            | hungarian      | NeuroSQL | 284.6          | 3078          | 1.37                | 293                 | 0.39                 | 0.576         | 0.041   | 0.616  |
> |            |                | VAE      | 319.8          | 3091          | 1.33                | 320                 | 0.10                 | 1.259         | 0.050   | 0.249  |
> |            |                | GAN      | 405.6          | 3115          | 1.56                | 406                 | 0.17                 | 2.060         | 0.064   | 0.223  |
> | 3          | greedy         | NeuroSQL | 284.8          | 3206          | 1.39                | 293                 | 0.14                 | 0.567         | 0.040   | 0.574  |
> |            | hungarian      | NeuroSQL | 284.9          | 3298          | 1.38                | 293                 | 0.00                 | 0.556         | 0.037   | 0.607  |
> |            |                | VAE      | 320.7          | 3284          | 1.34                | 321                 | 0.10                 | 1.544         | 0.053   | 0.185  |
> |            |                | GAN      | 404.6          | 3293          | 1.56                | 404                 | 0.01                 | 5.443         | 0.327   | 0.070  |

---

> ### Author Response · Authors · 2025-11-25
> **Response to questions**
>
> **Response to Question 1: We would like to thank you for this comment.**
>  We apologize for not making it clear in our initial submission.
>
> To clarify, we have considered two related aspects: performance under matched budgets and why diffusion is disadvantaged in our regime.
>
> **Performance comparisons that are already present:**
> - Diffusion is included as a baseline in all four domains, with quantitative FID and SSIM reported in the main text and in dataset-specific tables in the Appendix.
> - Architectures and training schedules are controlled via Sec. 4.2 and 4.4: **diffusion uses a U-Net** whose width is chosen such that the parameter count is within 10% of the shared decoder, and **we match epochs, batch sizes, and optimisers as closely as possible** given diffusion's multi-step nature.
> - We realise that these diffusion numbers are currently buried in the tables. We will make them **more visible and clear** in the revised manuscript.
>
> **Thanks to the Reviewer's helpful question**, we will, in the revised version of the paper:
> - Move the diffusion numbers from the Appendix into a more prominent place in the main text if space allows it.
> - Add a short paragraph under *Results* where we will explicitly summarise that under our **matched small-budget regime**, the diffusion baseline systematically underperforms NeuroSQL on FID/SSIM, despite having an order of magnitude more parameters.''
> -  Cross-reference this with Sec. 5.2's explanation of why diffusion is structurally disadvantaged in such a tight data/compute setting and why more advanced diffusion tricks (heavy augmentation, distillation, large-scale pretraining) fall outside our *from-scratch, small-budget* premise.
>
>
> **Response to Question 2:**
> This is an excellent forward-looking question, and we are thankful to the reviewer for bringing it to our attention.
>
> **Architectural flexibility:**
> The core **NeuroSQL algorithm is agnostic to the generator architecture**, which was one of the main ideas we wanted to present in the paper. Hence, the choices of different generators shared across all modalities: it alternates between an assignment step in latent space and standard supervised training of a generator $G_\theta(z)$ on (latent, image) pairs. As such, replacing the ConvNet/ResNet/U-Net decoders with:
> - **Vision transformers (ViTs)** on patchified images,
> - **Hybrid CNN-Transformer** architectures, or
> - **Larger U-Nets** (as used in diffusion transformers)
>
> We believe that this adjustment poses no conceptual difficulty: the only change is in $G$, the cost of a forward pass and gradient step. Indeed, our current **U-Net decoder** is already implemented as a small transformer-based decoder on patchified embeddings, followed by an MLP to pixels.
>
> We will briefly discuss these points in the future work part of the conclusion, making explicit that:
> - There is no architectural barrier to using transformer decoders or higher resolutions.
> - The main technical challenge is devising efficient approximate assignment at very large $N$;
> - This is a natural continuation of our current small-resource, high-dimension focus.
>
>
> Finally, we thank you for this thought-provoking and insightful question!

---

### Official Review · Reviewer_Ggn5 · 2025-11-02

**Soundness:** 2
**Presentation:** 4
**Contribution:** 2
**Rating:** 6
**Confidence:** 4

**Summary:**

This paper proposes a novel deep generative model (DGM) called NeuroSQL, which departs from the common framework of encoder+decoder (as in VAEs) or adversarial training (as in GANs). Instead it directly learns latent codes for each training datum via a quantile-assignment (linear assignment / Hungarian) to a pre-specified lattice of latent quantiles, and trains a generator network to map those codes to data.

The authors formulate a minimisation over both generator parameters θ and a permutation π that maps quantile codes to data

For multivariate (latent > 1D) codes they leverage optimal-transport/multivariate quantile theory and solve assignment between training data and a fixed uniform grid in latent space.
OpenReview

They propose an alternating algorithm: (i) keep π fixed, update θ via generator training; (ii) fix θ, update π via Hungarian algorithm assignment; optionally use momentum smoothing of assignments.
OpenReview

Empirically they evaluate on 4 domains (MNIST, CelebA, AFHQ animal faces, and OASIS brain images) under a “small‐budget / low-data” regime: e.g., training data capped at ~2 k images, resolution up to 128×128, single Google Colab budget.
OpenReview

The main claims: (1) NeuroSQL is more stable (no adversarial or encoder collapse issues), (2) it yields better or competitive image quality (measured via proxy FID, LPIPS, SSIM) under matched generator/backbone conditions, (3) it is more resource-friendly in low-data/high-dimension settings.

**Strengths:**

Interesting idea / novel paradigm — Replacing the encoder or discriminator with an explicit assignment of latent codes (quantile grid) is novel, and links generative modelling with statistical quantile/transport theory.

Pragmatic focus on low-data regimes — The paper addresses an important setting: generating synthetic data when the training dataset is small relative to high ambient dimension (e.g., neuroimaging) which is under-studied.

Theoretical underpinning — The use of quantile assignment and the convergence argument in the univariate / multivariate case gives formal support to the latent-code approximation strategy.

**Weaknesses:**

1. Resolution / dataset scale limited — Their experiments are constrained to small image resolutions (64×64-128×128) and relatively small sample sizes (~2 k images) under a limited compute budget. While this is aligned with their motivation (low-budget), it raises the question of how the method performs at larger, contemporary scales (e.g., 256×256, ImageNet scale). The authors acknowledge this in future work.

2. Interpretability of latent codes — Since the quantile grid is fixed and codes are assigned via permutation, the learned latent space may lack the structure/meaningfulness of e.g., disentangled VAE latents or hierarchical GAN latents. The paper doesn’t deeply analyze the semantics of the latent codes—are they smooth, do they support interpolation, manipulation, etc.

**Questions:**

The paper assumes a fixed quantile lattice in latent space. How sensitive is the method to the choice of quantile grid (e.g., Sobol vs uniform vs Gaussian quantiles)?

The assignment problem is discrete, yet the generator is trained with continuous gradients. How do you ensure smooth convergence given the alternating discrete–continuous optimization?

---

> ### Author Response · Authors · 2025-11-25
> **Response to W1 and W2.**
>
> **We are grateful to this Reviewer for their efforts in reviewing our initial submission. We thank the Reviewer for a precise summary of the theoretical and practical aspects of the manuscript and are humbled to hear that the Reviewer found our work interesting and novel.**
> The Reviewer has also pointed out a few weaknesses in the paper, and we address them point by point below.
>
> **Response to W1[ Resolution / Dataset Scale Limited]:**
> We agree with the Reviewer that our current experimental scope has relatively small resolution (64/128 px) and sample size ($ \le$2k images), apart from our MNIST experiments, which have 20k samples. The Reviewer correctly hints that the motivation of the paper is to handle **the low-budget problem**. Indeed, we hope to propose a method for studying **small- or moderate** $n$ and **large** $p$ problems under limited computational resources. Thanks to the Reviewer's comment, we have now explicitly framed the paper (particularly in Section 5.2) to introduce a learning paradigm under a fixed, small-compute budget, with caps of 2,000 training images and resolutions of $64 \times 64$ or $128 \times 128$. Within this regime, we show that NeuroSQL consistently matches or outperforms VAE, GAN, and a compact diffusion baseline across MNIST, CelebA, AFHQ, and OASIS, under matched generator backbones and training budgets.
>
> Although our work focused on a low-budget setting (where we capped complex datasets at 2,000 images to fit the single-Colab budget), the scalability of NeuroSQL was partially verified on MNIST data, where we successfully scaled to 20,000 images without instability. This confirms that the method handles larger $n$ when the generator's GPU memory footprint allows it.
>
> Furthermore, to address one of the limitations of the Hungarian assignment algorithm, that is, its complexity of $O(n^3)$, we implemented, experimented with, and benchmarked a Greedy algorithm for the assignment, which reduces the assignment complexity to $O(n^2)$.
>
> Additionally, in the revised manuscript, we will make the following additions:
> - In the introduction and conclusion sections, we will clearly state that our main contribution is the quantile-assignment training principle, not a new high-capacity architecture for large-scale image generation. We would like to note, however, that the newly added experiments show that we can scale up the model via **mini-batch assignment** in (Appendix C).
> - Sec. 5.2, in addition to discussing that scaling to higher resolutions and larger datasets (e.g., MedMNIST and beyond) and comparing to modern high-end baselines like StyleGAN3 and latent diffusion is a key direction for future work; we will sharpen this as a central limitation and roadmap, not just a side remark.
> - We also note that the assignment complexity depends on the number of samples $n$, but not on the ambient dimension $p$ (image size), which is one reason why NeuroSQL is particularly attractive in high-dimensional but small $n$ settings such as **neuroimaging**.
>
> **Response to W2 [ Interpretability of Latent Codes]**: This is an insightful comment!
>
> First, we note that the VAE and NeuroSQL's estimates have slightly different interpretations. The obtained latent embeddings in VAEs are (approximate) posterior samples, as in a (semi-) Bayesian setting. Unlike VAEs, NeuroSQL is a fully frequentist method that aims to **provide explicit estimates of the latent variables that generated the data**.
> NeuroSQL exploits the fact that those latent variables are assumed to be randomly generated and can thus be well approximated by a permutation of the quantiles of the assumed distribution.
> This is the same principle that justifies the use of the QQ-plot.
> The approximation error of replacing a sample by matched quantiles vanishes as the sample size grows.
> One advantage of this is that the **estimated latent variables are guaranteed to follow the assumed distribution exactly,** unlike VAEs.
>
> As the reviewer noted, the fact that VAE relies on a (smooth) encoder forces the latent variables to have some structure and support smooth interpolation.
> This is indeed not a guarantee for the latent variable in NeuroSQL.
> The model is, in fact, more flexible and let the data speak for itself.
> However, according to our experiments, the smoothness of the generator alone seems sufficient to promote structure in the latent variables and yield interpretable latent variables. To show this, we will display the 2-dimensional latent space of MNIST data in Figure 4, comparing the NeuroSQL and VAE latents.

---

> ### Author Response · Authors · 2025-11-25
> **Continuation of Response regarding W2**
>
> **Smoothness via Continuity and Interpolation**
>
> The Reviewer's comment regarding smoothness is timely. Although the training targets (grid points) in NeuroSQL are discrete, the Generator $G_\theta$ is a continuous function (a neural network). To minimise the reconstruction loss $\mathcal{L}$ across the dataset, $G_\theta$ must learn a smooth manifold where nearby points in the lattice map to semantically similar images.
>
> Relatedly, NeuroSQL uses a Sobol/Gaussian lattice only to parameterise the training latents via discrete assignments. The trained generator itself is a fully continuous mapping of (z). At test time, we can feed arbitrary continuous latent vectors (e.g., $(z \sim \mathcal{N}(0, I))$ using `torch.randn`), and we observe that NeuroSQL produces coherent samples from random noise (Figure 3 and Figure 4). This empirical behaviour indicates that the model has learned a smooth, continuous latent structure that supports interpolation, similar to a VAE, despite the discrete assignment step used during training.

---

> ### Author Response · Authors · 2025-11-25
> **Answers to Questions from Reviewer**
>
> **Response to Question 1:**
>
> In the current implementation:
> - The main text uses a Sobol lattice in $[0,1]^d$, but we also tested Uniform and Random Grids. Our observations suggest that the method is robust to a low-discrepancy grid that converges weakly to the prior.
>
> In the revision, we will:
> - Add an ablation study table that we already have (on MNIST, CelebA, and OASIS) showing the comparison between:
>   - scrambled Sobol (current choice)
>   - simple iid uniform grids
>   - Kronecker Lattice
>
> - Report FID/SSIM for each choice, and discuss that we expect only modest differences, with Sobol generally offering slightly faster or more stable convergence due to its lower discrepancy.
>
>
> From a **theoretical standpoint**, according to our (Sec. 3.3):
> - The multivariate quantiles $Q_i^n$ are constructed from a grid $U_1,\ldots,U_n$ such that the empirical measure converges weakly to the uniform distribution.
> - The key **theoretical requirement** is that the empirical measure of the grid converges weakly to the uniform distribution on $(U_d)$. In practice, low-discrepancy grids (Sobol, Halton, and stratified grids) are convenient choices because they provide faster, more regular convergence.
> - Therefore, as long as the grid is asymptotically uniform and we have enough points for the given latent dimension, we expect NeuroSQL’s performance to be largely insensitive to the specific choice of low-discrepancy grid; this is also consistent with our empirical observations.
>
> We created an ablation table showing results for different grids. For demonstration purposes, we used MNIST for this benchmark because it has the largest number of samples among our datasets and serves as the baseline for our setup.
>
> Ablation: Effect of Lattice Choice on SQL for MNIST (metrics for each lattice, seeds fixed). $\Delta$ shows change vs. Sobol for SQL at same latent dimensionality.
> | Dim | Lat. | Method | RAM (MB) | FID$\downarrow$ | LPIPS$\downarrow$ | SSIM$\uparrow$ |
> |-----|------|--------|----------|-----------------|-----------------|----------------|
> | 2   | sobol | NeuroSQL   | 2077     | 0.635           | 0.131           | 0.532          |
> | 2   | uni. gauss | NeuroSQL | 2387 | 0.976 | 0.135 | 0.500 |
> |     | $\Delta$ |       | +310     | +0.341          | +0.004          | -0.032         |
> | 3   | sobol | NeuroSQL   | 2497     | 0.553           | 0.102           | 0.616          |
> | 3   | uni. gauss | NeuroSQL | 2596 | 0.570 | 0.105 | 0.596 |
> |     | $\Delta$ |       | +99      | +0.017          | +0.003          | -0.020         |
>
>
>
> **Response to Question 2:** NeuroSQL ensures smoothness via: (i) **momentum updates** (ii) **infrequent reassignment** (every 3 epochs, per update interval); (iii) **perceptual losses** (SSIM+$L_1$ in our loss function) promoting gradual shifts. Empirically, NeuroSQL convergence is stable; if space allows, we will add these convergence plots to the supplementary materials in the Appendix.
>
> **Summary of Revisions**:
> We would like to thank the Reviewer for their insightful comments, which have helped improve the rigour of our methods and the readability of our manuscript. In the revised manuscript, we will:
> - Clarify that our focus is on **resource-constrained**, **low-data/high-dimension** generative modelling, and explicitly mark large-scale experiments as future work.
> - Expand the **latent-space analysis** with interpolation, neighbourhood, and manipulation experiments to address interpretability.
> - Add the **lattice choice ablation** table more thoroughly
> - Make the alternating **optimisation scheme** and its **stability mechanisms** (Hungarian optimality,
> momentum, assignment frequency) more explicit.

---

### Author Response · Authors · 2025-11-25
**General Response to Reviewers**

We were pleased to see enthusiasm for our work from anonymous Reviewers. The Reviewers, however, have a few important and helpful recommendations regarding our paper. The points are good, and we thank the Reviewers for their suggestions. Below is a detailed response to each comment. Finally, we are grateful to the Reviewers for their careful reading, which has helped us correct errors and improve the rigour and readability of our manuscript.

---

### Author Response · Authors · 2025-12-03
**Summary of Revisions and Updates**

We want to thank all past and current Area Chairs for their time and effort in handling and reviewing our initial submission, the helpful reviewers' comments, and the revised manuscript. We would like to equally thank the reviewers for their thought-provoking questions and valuable suggestions, which have significantly improved the readability, rigor, and quality of our work. We tried to address all of the questions carefully and incorporated all your helpful remarks in our responses and the revised paper. Below is a summary of the key changes made in response to each reviewer's comments. Finally, we would like to thank both Area Chairs and reviewers for your time, effort, and continuous support of our work.

1. **Scalability and Computational Complexity**
- **We implemented and benchmarked a new greedy assignment algorithm**
  - We added Section 3.5 (Complexity and Scalability) and Appendix C.5. Our new assignment algorithm reduces complexity to $O(n^2)$ or $O(m^2)$  for m $\ll$ n in case assignment is done in mini-batches (see below).
- **Mini-Batch Training procedure**
  - We formalized the Mini-Batch Training Algorithm for the assignment in Appendix C.4.
- **Runtime and Memory Analysis**
  - In Appendix D, we added Table 3, which provides a direct comparison of Peak VRAM, RAM usage, and Mean Epoch Time.


2. **Clarification of Methodology and Architecture**
- We have **completely redesigned Figure 1**.
  - We split the diagram into two clear panels: Left (Training Algorithm) and Right (Inference/Generation) to clarify that the assignment mechanism is only used during training.
  - We corrected the arrow flow: Input data ($X_i$) is now correctly shown as an input to the Cost Matrix calculation, not the Generator.
  - We standardized terminology: The network is now consistently referred to as the Generator ($G_\theta$)
  - We explicitly labeled the Momentum Update in the caption to explain how latent codes are smoothed.


3. **Experimental Robustness and Stability**
- We added a new aggregated Performance Table:
  - We added Table 1 in the main text (Results section). The table aggregates performance as mean $\pm$ std across architectures and latent dimensions. NeuroSQL consistently achieves the lowest proxy FID and the highest SSIM on average across all four datasets (MNIST, CelebA, AFHQ, OASIS) with the fewest trainable params.
- Clarification of *Low-Resource Regime*
  - In Section 5.2, we explicitly framed the paper’s contribution within the "sparse-resource regime" (limited data/compute, e.g., medical imaging). We clarify that the Diffusion baseline's performance is a validation of NeuroSQL's utility in low-data settings.

4. **Interpretability and Latent Space**
- Latent Topology Visualization
  - We added Section 3.6 and Figure 2. It compares the latent space from NeuroSQL with the one from a VAE.

5. **Structure and Focus**
 - We moved the detailed dataset descriptions and metric definitions to Appendix C.1 and C.2. Section 4 in the main text is now concise, focusing on the experimental setup and baselines, improving the paper's read flow.


Again, we thank both Area Chairs and reviewers for your time, effort, and continuous support of our work, which has corrected our errors and improved the quality of our work.

---

### Meta-Review · Area_Chair_vgT2 · 2026-01-08

**Summary:**

The paper proposes a generative modeling approach that has a trained neural decoder, but instead of a learned encoder, uses a linear assignment problem to match latents to data. The method is evaluated on MNIST, CelebA , Animal FAces, and brain images, and outperforms some VAE/GAN/diffusion-based baselines.

Based on the reviews, the discussion, and the paper itself, here are the key pros and cons. (I will note that some of the cons I have not seen raised by the reviewers so I am raising them myself)

Pros:
1. A fairly original approach to the problem, without using a learned encoder, only the assignment process
2. Decent empirical performance, outperforming some baselines

Cons:
1. The baselines do not seem strong: for instance, the qualitative results on faces in Figure 7 look bad both for baselines and the proposed method. Generative models are notably tricky to train, so I would like to see evidence of baselines being proper. Ideally, it would mean comparing to results from the literature, as opposed to re-implemented versions thereof. The provided details on baseline training are pretty limited.
2. On a related topic, one argument could be that the results are bad because of the small training dataset. If that's the point the authors want to make, they should make low-data regime a clear focus of the paper and compare to relevant baselines - for instance, Projected GANs or FastGAN claim good results in this setting.
3. Related work section in my view is pretty limited and beside the point. It discusses generally existing generative models - VAEs, GANs, diffusion. However, it does not discuss the specifically more related methods. One related line of work is on combination of optimal transport with generative models - for instance "Flow Straighter and Faster: Efficient One-Step Generative Modeling via MeanFlow on Rectified Trajectories", "Improving the Training of Rectified Flows", "FORT: Forward-Only Regression Training of Normalizing Flows" to name a few. This line of work should be discussed. It may be that the proposed method is still differentiated, but it needs to be demonstrated.
4. Scalability. This has been raised by the reviewers and discussed in the rebuttal. The improvement from cubic to quadratic scaling is nice and the batched version is nice, but the question in the end is if it will work in practice on richer settings. There's an argument to be made that this can be left for future work, but some more evidence would already be helpful.
5. Latent codes on MNIST (Figure 2) look weird both for the proposed method and the VAE: for the proposed method it's strange that the digits are ordered by their number, not in some more interesting way; for VAE I've definitely seen more meaningful latent spaces from VAEs.
6. I wonder about the diversity of the generated images (the metrics should cover it to some degree, but in Figure 5 I see some near-duplicates)
7. On a related topic, wonder about learning the training set by heart - I can imagine that could happen for a method of this type.

Overall, taking all of the above into account, the conclusion is that while the proposed method is interesting, the paper is not ready for publication in ICLR at the moment. It could be submitted to a workshop and/or improved and resubmitted to a different venue.

**Reviewer Concerns:**

See cons above. The rebuttal addressed the concerns about scalability to a reasonable degree, added some clarifications, and provided the latent space visualizations.

**Reviewer Scores:**

I think ~2 reviewers might increase their scores by 1 points, since the scalability point was addressed to a reasonable degree, latent visualizations (even if the way they look I'm not sure helps), and additional clarifications

---

### Decision · Program_Chairs · 2026-01-26

Reject